# LiDAR-Anchored Collaborative Distillation for Robust 2D Representations

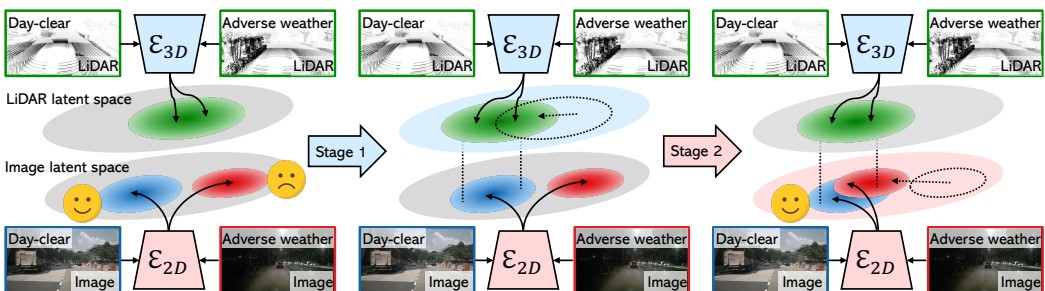

Figure 1: **Collaborative Distillation.** Under adverse weather conditions, the 2D feature distribution degrades (red) while the 3D feature distribution remains stable (green). Stage 1 aligns the 3D feature distribution to the 2D clear-side (blue). Stage 2 uses the aligned 3D features to pull the 2D degraded-side toward the 2D clear-side. This yields robust 2D features with original semantic context.

## Abstract

As deep learning continues to advance, self-supervised learning has made considerable strides. It allows 2D image encoders to extract useful features for various downstream tasks, including those related to vision-based systems. Nevertheless, pre-trained 2D image encoders fall short in conducting the task under noisy and adverse weather conditions beyond clear daytime scenes, which require for robust visual perception. To address these issues, we propose a novel self-supervised approach, **Collaborative Distillation**, which leverages 3D LiDAR as self-supervision to improve robustness to noisy and adverse weather conditions in 2D image encoders while retaining their original capabilities. Our method outperforms competing methods in various downstream tasks across diverse conditions and exhibits strong generalization ability. In addition, our method also improves 3D awareness stemming from LiDAR's characteristics. This advancement highlights our method's practicality and adaptability in real-world scenarios. The code will be released upon acceptance.

## 1 Introduction

While deep learning models have shown considerable progress (Chen et al., 2018b; Kirillov et al., 2023; Zou et al., 2024), many of them rely on supervised learning, which requires extensive human labeling, a costly process (Zou et al., 2020; Genova et al., 2021). In contrast, self-supervised learning methods, which are label-efficient, have shown significant progress in the image domain (Chen et al., 2020a; Caron et al., 2021; Zhou et al., 2022; Oquab et al., 2024; Siméoni et al., 2025). These self-supervised learning methods allow the image encoders to learn versatile features that are effective for downstream tasks, such as semantic segmentation and depth estimation, which are beneficial in vision-based systems (Chen et al., 2021a; Guizilini et al., 2020).

However, the pre-trained models obtained by these self-supervised learning approaches often face a challenge: poor robustness to noisy and adverse weather conditions, such as low-light, rainy, or corrupted conditions. In real-world scenarios, vision-based systems must operate reliably under these challenging conditions. Pre-trained image encoders often struggle in such environments, likely

because their pre-training datasets, such as ImageNet-1K (Deng et al., 2009), largely consist of clear daytime images, as well as limitations arising from the self-supervision loss. This raises a key question: can pre-trained image encoders overcome these challenges by learning from reliable self-supervision, which is not easily available from noisy and adverse 2D image data alone?

In this context, we leverage 3D LiDAR data, which offers two advantages: (1) greater robustness to adverse weather conditions compared to cameras (Kim et al., 2023) and (2) inherent 3D information, both of which 2D images generally lack. It can offer a reliable self-supervision under adverse weather conditions. The question is how to enable a 2D image encoder to learn these favorable properties from 3D LiDAR data effectively. Since raw 3D LiDAR data lacks semantic context, directly injecting this information may degrade the 2D encoder's original capabilities. To prevent this degradation, we propose a two-stage self-supervised distillation method (see Fig. 1), *collaborative distillation*, to preserve the original strengths of 2D encoders while enabling them to receive reliable self-supervision under noisy and adverse weather conditions, thereby enhancing robustness to noisy and adverse weather conditions. We demonstrate the effectiveness of our method on the downstream tasks, including depth estimation, semantic, and depth-aware video panoptic segmentation, using both in-domain and out-of-domain datasets. We summarize our main contributions as follows:

- To the best of our knowledge, this study is the first self-supervised representation learning to enhance the robustness of 2D image encoders under noisy and adverse weather conditions.
- We propose a collaborative distillation method that effectively exchanges the complementary properties of multi-modal data (images and 3D LiDAR points) and preserves their original capabilities.
- Our method enhances 2D image encoders across domains, demonstrating strong generalization and adaptability, and also improves 3D awareness, which is beneficial for vision-based systems.

## 2 RELATED WORK

Our work is related to 2D image and 3D LiDAR self-supervised representation learning. We focus on improving 2D representations in adverse conditions, with 3D awareness as a by-product. Therefore, we brief the related work: 2D self-supervised representation learning, image-to-LiDAR distillation, improving 2D encoders in low-visibility scenarios, and improving 3D awareness of 2D representation.

**2D self-supervised representation learning.** Visual self-supervised representation learning aims to learn visual features, which can be applied to various downstream tasks (Liu et al., 2021). Recently, three pretext tasks are commonly chosen as main strategies: (1) contrastive learning, where a 2D encoder learns to extract features that are invariant to augmentations (Chen et al., 2020a; He et al., 2020; Henaff, 2020; Misra & Maaten, 2020; Oord et al., 2018; Tian et al., 2019; Wu et al., 2018; Chen et al., 2020b); (2) masked image modeling, where the encoder is trained to reconstruct masked parts of an image (Bao et al., 2021; He et al., 2022); and (3) self-distillation, where a student model learns by predicting the features of a teacher model (Caron et al., 2021; Zhou et al., 2022; Oquab et al., 2024; Siméoni et al., 2025). However, these methods are trained mostly on clear daytime images, leaving their effectiveness in noisy and adverse weather conditions underexplored, posing a key challenge: extracting robust 2D representations. For the first time, our method tackles this challenge from a representation learning perspective via self-supervised distillation approach.

**Image-to-LiDAR distillation.** While traditional 3D self-supervised approaches learn only from 3D LiDAR point clouds (Chen et al., 2021b; Sauder & Sievers, 2019; Wang et al., 2021a; Hou et al., 2021a; Xie et al., 2020; Nunes et al., 2022; Zhang et al., 2021; Yin et al., 2022), recent strategies also leverage the semantic information of the image domain by pre-training 3D encoders on image-LiDAR pairs. SLidR (Sautier et al., 2022) pioneered an image-to-LiDAR distillation, where 2D image superpixels and matched 3D point clouds are compared through contrastive learning. Building on SLidR, several methods (Mahmoud et al., 2023; Pang et al., 2023; Liu et al., 2023; Chen et al., 2023; Xu et al., 2024; 2025) develop losses to push performance further or revisit the data utilization (Jo et al., 2024). OccFeat (Sirko-Galouchenko et al., 2024) also distills image features into point clouds, but its goal is to improve multi-camera BEV perception. ScaLR (Puy et al., 2024) scaled up the dataset and the model size. These methods are based on the pre-trained image encoder, and target to learning 3D representations only. In contrast, we leverage 3D LiDAR representations, which are highly robust to adverse weather conditions, to enhance 2D representations. We employ this image-to-LiDAR distillation scheme to perform Stage 1.

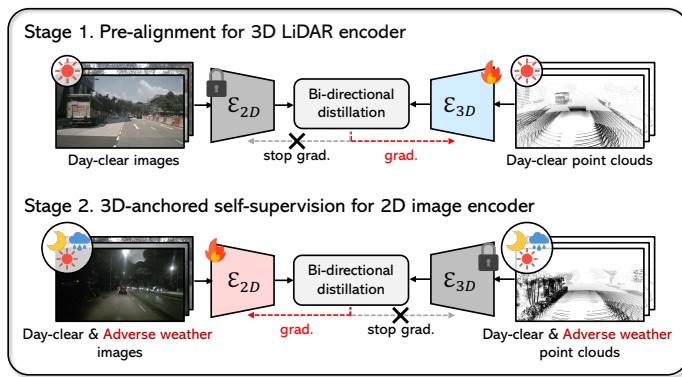 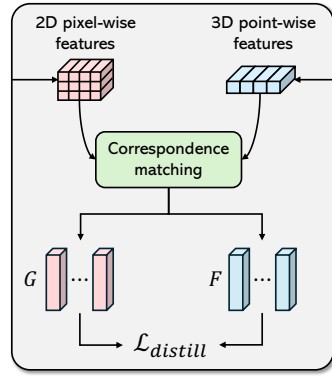

(a) Collaborative distillation framework       (b) Bi-directional distillation module

Figure 2: **Overall Pipeline of the proposed method**. (a) Stage 1 (Pre-alignment) aligns the 3D features to the clear-side 2D features, and Stage 2 (3D-anchored self-supervision) pulls degraded 2D features under adverse conditions toward the pre-aligned 3D features. (b) The bi-directional distillation module matches pixel- and point-wise features and applies cross-modal distillation loss.

**Improving 2D encoders in low-visibility scenarios.** Deficient robustness of 2D encoders under low-visibility or adverse weather conditions is a growing concern for real-world applications. There are two streams in this field. Many studies focus on improving visibility under nighttime, rainy, or foggy conditions, so that images become more visually clear (*e.g.*, de-rain, de-fog, low-light enhancement) (Chen et al., 2018a; Jiang et al., 2021; Wang et al., 2022; Lore et al., 2017; Lv et al., 2018; Peng et al., 2024; Qian et al., 2024; Yang et al., 2024a). However, these methods primarily improve image quality and do not enhance representations for downstream tasks. Another line of research focuses on task-specific robust recognition under low-visibility or adverse weather (Gasperini et al., 2023b; Wang et al., 2021b; Lee et al., 2023; Sasagawa & Nagahara, 2020; Spencer et al., 2020; Saunders et al., 2023; Gasperini et al., 2023a) , including cross-modal unsupervised domain adaptation methods (Jaritz et al., 2020; 2022). In contrast to these studies that are limited to the specific task (*e.g.*, depth estimation, object detection, pose estimation, or semantic segmentation), our task-agnostic, self-supervised distillation improves the representation of the 2D encoders for low-visibility/adverse-weather and generalizes across tasks.

**Improving 3D awareness of 2D representation.** There have been studies aimed at improving the 3D awareness of 2D representations. Various methods (Hou et al., 2021b; Bachmann et al., 2022; Hou et al., 2023; Weinzaepfel et al., 2022) incorporate 3D priors through multi-view geometry or masked image modeling with RGB-D. Recently, FiT3D (Yue et al., 2024) lifts 2D features into 3D Gaussians and applies multi-view rendering. Condense (Zhang et al., 2024) enforces 2D-3D feature via ray marching (Mildenhall et al., 2020), enabling training of 2D and 3D encoders. Unlike the above studies that focus on 3D priors in the indoor scenarios, several studies (Hong et al., 2022; Chong et al., 2022; Chen et al., 2022; Wang et al., 2023; Li et al., 2022; Yang et al., 2024b; Kim et al., 2024; Klingner et al., 2023; Sun et al., 2024) have utilized cross-modal distillation between LiDAR and images in the outdoor scenarios, targeting a specific task, *e.g.*, 3D or BEV-based object detection. In contrast, we aim to enhance the representation of the 2D encoders that take a 2D image as input, enabling them to perform various downstream tasks in and out-of-domain for broad applicability.

## 3 COLLABORATIVE DISTILLATION

In this section, we introduce a self-supervised collaborative distillation method. This method is divided into stages 1 and 2 (see Fig. 2), which we describe sequentially in Secs. 3.1 and 3.2.

### 3.1 STAGE 1: PRE-ALIGNMENT

The goal of Stage 1 is to pre-align the LiDAR encoder's features to the 2D encoder's clear-side features. To perform Stage 1, we prepare the pre-trained 2D encoder $\mathcal{E}_{2D}$, followed by a bilinear upsampling layer to restore the reduced feature map to the original image resolution. For the 3D part,

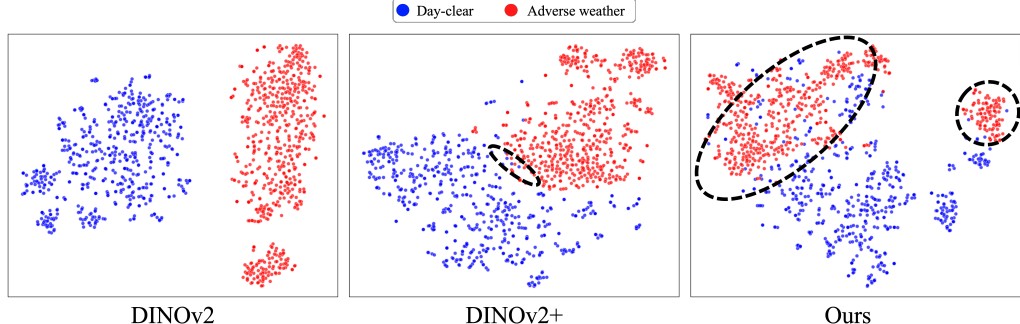

Figure 3: **t-SNE visualization of extracted image features.** Compared with DINOv2 and DINOv2+[1], where clear- and adverse-side clusters remain separated, our method shifts adverse-side toward the clear-side cluster, achieving the intended distribution shift.

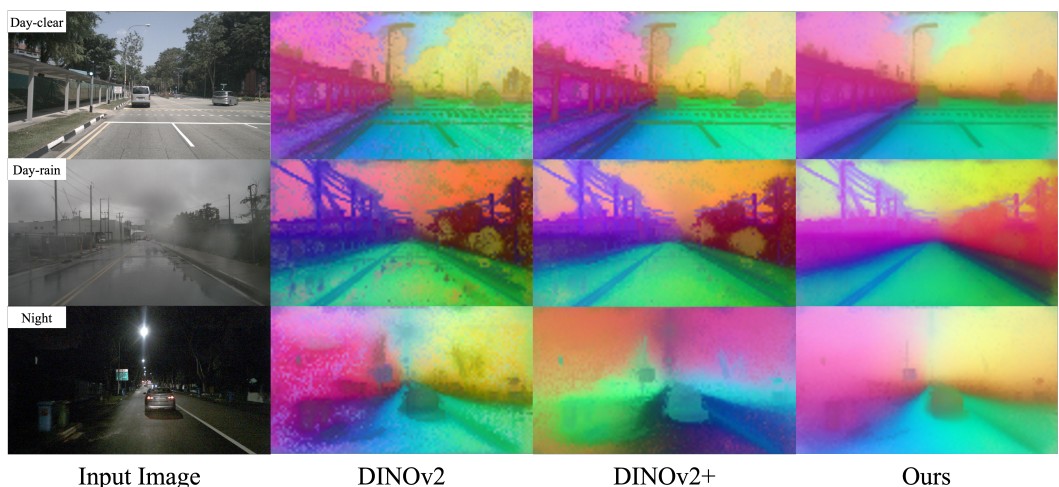

Figure 4: **Feature Visualization.** Compared with DINOv2 and DINOv2+, our method produces cleaner feature across all conditions, indicating improved robustness and feature denoising effect.

we prepare the 3D encoder $\mathcal{E}_{3D}$, followed by a 3D linear head $\mathcal{H}_{3D}$ to align the 3D feature dimension to the 2D one. For simplicity, we omit to note the bilinear upsampling layer and the 3D linear head.

**Assumption 1 (Day-clear reliability).** We assume that, because the 2D encoder $\mathcal{E}_{2D}$ was predominantly pre-trained on daylight images, it yields reliable features mainly on day-clear inputs; thus we use only day-clear image–LiDAR pairs in Stage 1.

**Description of Stage 1.** Let $P = [\mathbf{p}_1, ..., \mathbf{p}_N]$ and $I \in \mathbb{R}^{H \times W \times 3}$ denote 3D LiDAR point cloud and 2D image, where $\mathbf{p}_i \in \mathbb{R}^3$ is the $i$-th 3D point, and $H$, $W$ and $N$ are the height and width of the image, and the number of points, respectively. We extract pixel-wise features from the 2D encoder $\mathcal{E}_{2D}$ and point-wise features from the 3D encoder $\mathcal{E}_{3D}$. Within the Bi-directional distillation module, we obtain $M$ pairs of features $\mathbf{G} = [\mathbf{g}_1, ..., \mathbf{g}_M]$ and $\mathbf{F} = [\mathbf{f}_1, ..., \mathbf{f}_M]$ through correspondence matching, where $\mathbf{g}_i, \mathbf{f}_i \in \mathbb{R}^D$ denote the $i$-th matched pixel and point features. For each pair $\{\mathbf{g}_i, \mathbf{f}_i\}$, we apply an image-to-LiDAR distillation loss to make $\mathbf{f}_i$ similar to $\mathbf{g}_i$, defined as follows:

$$\mathcal{L}_{\text{distill}} = \frac{1}{M} \sum\nolimits_{i \in M} \|\text{sg}[\mathbf{g}_i] - \mathbf{f}_i\|_2 , \tag{1}$$

where $\mathbf{f}_i$ and $\mathbf{g}_i$ are $l_2$-normalized and $\text{sg}[\cdot]$ stands for stop-gradient operator. Through the Stage 1, we obtain a $\mathbf{F}$ aligned to the clear-side $\mathbf{G}$, making $\mathbf{F}$ a reliable anchor for self-supervision in Stage 2.

---

[1] DINOv2 further trained on the nuScenes (Caesar et al., 2020) dataset including adverse conditions, using the same training protocol as DINOv2.

| Method | Arch. | nuScenes | | | | | | | | nuImages | |
|---|---|---|---|---|---|---|---|---|---|---|---|
| | | full | | day-clear | | day-rain | | night | | full | |
| | | 1% FT | 100% LP | 1% FT | 100% LP | 1% FT | 100% LP | 1% FT | 100% LP | 1% FT | 100% LP |
| DINOv2 | ViT-S/14 | 35.2 | 49.2 | 36.2 | 50.5 | 33.9 | 46.8 | 22.4 | 27.7 | 63.2 | 70.9 |
| + CD | | 35.7 (+0.5) | 51.9 (+2.7) | 36.8 (+0.6) | 52.7 (+2.2) | 34.2 (+0.3) | 49.8 (+3.0) | 23.9 (+1.5) | 33.4 (+5.7) | 64.0 (+0.8) | 71.7 (+0.8) |
| DINOv2 | ViT-B/14 | 39.0 | 52.3 | 40.5 | 53.3 | 38.1 | 50.1 | 24.5 | 33.8 | 70.4 | 74.9 |
| + CD | | 39.8 (+0.8) | 55.5 (+3.2) | 41.1 (+0.6) | 56.6 (+3.3) | 38.5 (+0.4) | 54.7 (+4.6) | 26.5 (+2.0) | 37.4 (+3.6) | 70.7 (+0.3) | 76.4 (+1.5) |
| DINOv2 | ViT-L/14 | 42.7 | 53.6 | 44.0 | 54.2 | 39.8 | 52.8 | 26.8 | 36.5 | 73.1 | 75.7 |
| + CD | | 43.7 (+1.0) | 57.6 (+4.0) | 44.9 (+0.9) | 58.1 (+3.9) | 40.7 (+0.9) | 56.8 (+4.0) | 30.7 (+3.9) | 42.0 (+5.5) | 74.4 (+1.3) | 77.9 (+2.2) |
| DINOv2 | ViT-G/14 | 44.4 | 55.1 | 45.8 | 55.8 | 42.2 | 53.9 | 29.4 | 37.6 | 75.0 | 77.7 |
| + CD | | 47.1 (+2.7) | 58.8 (+3.7) | 48.3 (+2.5) | 59.3 (+3.5) | 43.6 (+1.4) | 57.8 (+3.9) | 33.3 (+3.9) | 43.0 (+5.4) | 75.8 (+0.8) | 79.4 (+1.7) |

Table 1: **In-domain linear probing (LP) and few-shot fine-tuning (FT) performance for 2D semantic segmentation.** Our method consistently improves mIoU across all conditions, with larger gains under day-rain and night, indicating improved robustness.

**Correspondence matching.** Given a calibrated relative pose between the LiDAR and the camera, the 3D-to-2D projection $\mathcal{T} : \mathbb{R}^3 \rightarrow \mathbb{R}^2$ outputs the projected 2D coordinate on the image $I$, *i.e.*, $\mathbf{x}_i = \mathcal{T}(\mathbf{p}_i)$. Then, we collect the $M$ pixel indices from all visible $\mathbf{x}_i$ in the image. Since pixel and point indices are paired by $\mathcal{T}$, we use these pairs to retrieve the corresponding pixel-wise and point-wise features. Finally, we obtain the $M$ pixel-point feature pairs $\{\mathbf{g}_i, \mathbf{f}_i\}$.

### 3.2 STAGE 2: 3D-ANCHORED SELF-SUPERVISION

The goal of Stage 2 is to pull the degraded $\mathbf{G}$ under adverse conditions toward the clear-side $\mathbf{G}$ for robustness, using the pre-aligned $\mathbf{F}$ as a 3D-anchored self-supervision. To perform Stage 2, we continue to use the pre-trained 2D encoder $\mathcal{E}_{2D}$ and the 3D encoder $\mathcal{E}_{3D}$ trained in Stage 1. At this stage, we use all data including adverse weather conditions.

**Assumption 2 (3D-anchored stability).** We assume that LiDAR is less affected by adverse weather, so its data distribution stays close to day-clear and so does its feature distribution; Thus we use the pre-aligned $\mathbf{F}$ under adverse conditions as reliable 3D-anchored self-supervision in Stage 2.

**Description of Stage 2.** We implement the Stage 2 by switching the gradient direction, reusing the Bi-directional distillation module and the weights of the 2D encoder $\mathcal{E}_{2D}$ and 3D encoder $\mathcal{E}_{3D}$ from Stage 1 without adding any additional layer. To preserve the 2D encoder's original capabilities and provide reliable self-supervision, it is crucial to keep the Stage 1 loss and weights unchanged and switch only the gradient direction. The Stage 2 loss is defined as follows:

$$\mathcal{L}_{\text{distill}} = \frac{1}{M} \sum_{i \in M} \| \mathbf{sg}[\mathbf{f}_i] - \mathbf{g}_i \|_2 , \tag{2}$$

where $\mathbf{f}_i$ and $\mathbf{g}_i$ are $l_2$-normalized. Through Stage 2, we expect the further trained 2D encoder $\mathcal{E}_{2D}$ to extract robust 2D features $\mathbf{G}$, while preserving its semantic context established in Stage 1.

**Emergent effects of Collaborative Distillation.** After Collaborative Distillation, we examine how our method changes the 2D features. **(i) t-SNE shift toward clear-side.** In DINOv2 and DINOv2+, clear-side and adverse clusters are largely separate; with ours, adverse-side shift to overlap the clear-side cluster (see Fig. 3), exhibiting the intended distribution shift. **(ii) Denoising in feature maps.** Compared with DINOv2 and DINOv2+, our method yields noticeably cleaner feature maps under adverse conditions (see Fig. 4), indicating improved robustness and an emergent feature denoising effect. Detailed visualization procedures are in the Appendix.

## 4 EXPERIMENTS

We first describe the experimental setup (Sec. 4.1), and then present results for in-domain and out-of-domain downstream tasks (Secs. 4.2 and 4.3). Finally, we provide ablation studies (Sec. 4.4).

### 4.1 EXPERIMENTAL SETUP

**Encoders.** We use the WaffleIron-768 (Puy et al., 2023) as a 3D encoder and various pre-trained 2D encoders. Our primary models are ViT-S/14 to ViT-G/14, pre-trained by the DINOv2 (Oquab et al., 2024). The above pre-trained models are linear-probed or fine-tuned on downstream tasks.

**Datasets.** We pre-train all models on the nuScenes dataset (Caesar et al., 2020), which contains 168k/36k images (train/val). Of these, 20k/3.6k images are night, and 32.8k/6.0k are rainy. For in-domain experiments, nuScenes and nuImages (Caesar et al., 2020) datasets are leveraged. For out-of-domain experiments, we follow the protocol of DINOv2 (Oquab et al., 2024) with out-of-domain datasets such as KITTI (Geiger et al., 2012), NYUd (Silberman et al., 2012), Cityscapes (Cordts et al., 2016), and ADE20k (Zhou et al., 2017). For the multi-task experiment in night and rainy scenarios, we convert all test day images of the Cityscapes-DVPS (Qiao et al., 2021) into night and rainy images using the Stable Diffusion (Rombach et al., 2022)-based image translation method (Parmar et al., 2024). For the image corruption experiment, we convert all nuScenes test images into fog, Gaussian noise, and left-right motion blur using the corruption algorithms of Dong et al. (2023b).

**Data augmentation.** For the 2D encoders, we resize images to $224 \times 448$ in all stages. For the 3D encoder, we apply random z-axis rotation and xy-axis flipping in Stage 1.

**Hyperparameters.** For Stage 1, we pre-train the WaffleIron (Puy et al., 2023) encoder using the AdamW (Loshchilov, 2017) optimizer, setting weight decay to $3 \times 10^{-2}$ and a learning rate that starts at 0, increases to $2 \times 10^{-3}$, and then decreases to $10^{-5}$ following a cosine schedule. The batch size is 8, and we train for 49 epochs. For Stage 2, the hyperparameters are nearly identical to those in Stage 1, with adjustments to batch size, learning rate, and epochs. The batch size is 32, with learning rates of $2 \times 10^{-5}$ for ViT and $5 \times 10^{-3}$ for ResNet50. The training epoch is set to 1. All pre-training is conducted on 4 NVIDIA A100 GPUs.

## 4.2 TRANSFER TO IN-DOMAIN DOWNSTREAM TASKS

We verify whether our method improves robustness on the validation set of the pre-training dataset *i.e.*, in-domain. Therefore, we compare the models pre-trained by DINOv2 with the models after Collaborative Distillation (CD) on in-domain downstream tasks, specifically semantic segmentation on both nuScenes (Caesar et al., 2020) and nuImages (Caesar et al., 2020), and depth estimation on nuScenes. Since the nuScenes dataset lacks 2D semantic segmentation labels, we project the 3D LiDAR semantic segmentation labels onto the images as ground truth. We use the entire training set for downstream tasks, and for the validation set, we split into day-clear, day-rain, and night to check the model's robustness. Each 2D task-specific linear head replaces the linear head used during distillation, mapping pixel-wise features to segmentation classes or depth values.

For semantic segmentation, we conduct $1\%$ label fine-tuning to assess effectiveness in label-scarce scenarios and use full-label linear probing to evaluate the effectiveness of the learned representations. We follow the above training and evaluation protocol of (Puy et al., 2024). Similarly, for depth estimation, we perform linear probing with all labels to evaluate the learned representations and further fine-tune with all labels to assess how closely the model approaches state-of-the-art robust depth estimation method (Gasperini et al., 2023b). We measure mean Intersection over Union (mIoU) for semantic segmentation and Root Mean Squared Error (RMSE) for depth estimation.

**Semantic segmentation.** Table 1 shows that our method outperforms all mIoU metrics, with linear probing on all labels and fine-tuning with 1%. Interestingly, our method is effective in both label-scarce and full-label evaluations while improving performance across all metrics on day-clear images. We assume that by incorporating LiDAR properties, the 2D representation preserves its semantics while becoming more discriminative, as it integrates 3D depth information to differentiate objects with similar appearances but different spatial depths (Li et al., 2024; Ji-Yeon et al., 2024; Ye & Xu, 2023). Additionally, the improvement is larger at day-rain and night than during the day-clear, indicating that our method reduces the domain gap, enhancing the 2D encoder's robustness.

**Depth estimation.** Table 2 shows the fine-tuning and linear probing RMSE results using all labels on nuScenes. Our method demonstrates consistent improvements across all scenarios, which indicates that our method learns robust 2D representations. These results also suggest that LiDAR's geometric cues are effectively distilled into the 2D encoder, thereby enhancing its 3D awareness. Although we use only a simple linear layer and MSE loss for depth estimation training, our method achieves comparable performance to state-of-the-art approaches (Gasperini et al., 2023b) focused on robust depth estimation relying on advanced methods (bha, 2021).

| Method | Arch. | full (-) | | day-clear (4.81‡) | | day-rain (5.90‡) | | night (6.37‡) | |
|---|---|---|---|---|---|---|---|---|---|
| | | 100% FT | 100% LP | 100% FT | 100% LP | 100% FT | 100% LP | 100% FT | 100% LP |
| DINOv2 | ViT-S/14 | 5.72 | 8.37 | 5.46 | 8.14 | 6.00 | 8.79 | 7.00 | 9.40 |
| + CD | | **5.70** | **7.64** | **5.43** | **7.47** | **5.97** | **7.66** | **6.98** | **8.73** |
| DINOv2 | ViT-B/14 | 5.46 | 8.01 | 5.20 | 7.86 | 5.62 | 8.26 | 6.87 | 8.76 |
| + CD | | **5.42** | **7.18** | **5.17** | **7.01** | **5.58** | **7.19** | **6.76** | **8.28** |
| DINOv2 | ViT-L/14 | 5.29 | 7.97 | 5.00 | 7.81 | 5.38 | 8.32 | 6.57 | 8.69 |
| + CD | | **5.22** | **6.96** | **4.98** | **6.81** | **5.31** | **7.08** | **6.55** | **7.82** |
| DINOv2 | ViT-G/14 | 5.18 | 7.66 | 4.93 | 7.49 | 5.27 | 8.09 | 6.55 | 8.33 |
| + CD | | **5.14** | **6.63** | **4.91** | **6.47** | **5.22** | **6.89** | **6.49** | **7.40** |

Table 2: **In-domain linear probing and fine-tuning performance for 2D monocular depth estimation.** Our method consistently improves RMSE across all conditions, indicating robust 2D representations and improved 3D awareness from complementary LiDAR signals. The state-of-the-art performances from md4all (Gasperini et al., 2023b) are denoted at the top of the table with ‡.

| Method | Arch. | Depth (RMSE ↓) | | Seg. (mIoU ↑) | |
|---|---|---|---|---|---|
| | | KITTI | NYUd | Cityscapes | ADE20k |
| DINOv2 | | 2.86 | 0.397 | 73.7 | 47.2 |
| + FiT3D[†] | ViT-B/14 | 2.79 | 0.380 | - | **49.5** |
| + CD | | **2.63** | **0.376** | **74.0** | 47.9 |
| DINOv2 | | 2.60 | 0.339 | 73.6 | 47.9 |
| + Condense | ViT-G/14 | 2.67 | 0.320 | 74.1 | 48.7 |
| + CD | | **2.46** | **0.319** | **75.6** | **51.0** |

Table 3: **OOD linear probing for 2D monocular depth estimation and semantic segmentation.**

| Method | Eval. data | Video panoptic seg. | Depth |
|---|---|---|---|
| | | VPQ ↑ | RMSE ↓ |
| DINO | Rainy | 25.8 | 6.78 |
| + CD | | **29.7** | **6.17** |
| DINO | Night | 33.7 | 5.23 |
| + CD | | **34.2** | **4.42** |

Table 4: **OOD multi-task linear probing results under adverse weather.**

## 4.3 Transfer to Out-of-Domain Downstream Tasks

In this section, we demonstrate that our method generalizes well to out-of-domain (OOD) downstream tasks, including indoor environments, even though it is pre-trained soley on a nuScenes for outdoor autonomous driving. Beyond assessing out-of-domain generalization, we also examine improvements in 3D awareness. Therefore, we compare the models pre-trained by DINOv2, two 3D prior injection methods (FiT3D (Yue et al., 2024) and Condense (Zhang et al., 2024)), and our CD method on out-of-domain downstream tasks. All training and evaluation protocols follow the DINOv2 setup. Depth estimation is trained and evaluated on the outdoor KITTI (Geiger et al., 2012) and indoor NYUd (Silberman et al., 2012) datasets, while semantic segmentation is conducted on outdoor Cityscapes (Cordts et al., 2016) and ADE20k (Zhou et al., 2017), which consists of indoor and outdoor data. Note that FiT3D combines the model pre-trained by DINOv2 and the model improved by their method for downstream tasks (*i.e.*, assembling). † in Table 3 indicates the use of the assembling technique. In addition, we demonstrate the effectiveness of our 2D encoder in a multi-task learning setup, which jointly handles video panoptic segmentation and depth estimation. We adopt the recent framework (Ji-Yeon et al., 2024), replacing its image encoder with ours. We train and validate the model with Cityscapes-DVPS (Qiao et al., 2021) datasets.

**Depth estimation.** Table 3 shows the linear-probing RMSE results on the KITTI (Geiger et al., 2012) and NYUd (Silberman et al., 2012) datasets. Our method achieves superior results across all metrics on both outdoor and indoor datasets (KITTI and NYUd). Considering our method is pre-trained solely on the outdoor dataset, nuScenes, this generalization ability is noteworthy. Note that FiT3D and Condense are pre-trained on indoor datasets (Yeshwanth et al., 2023; Dai et al., 2017; Chang et al., 2015), and FiT3D employs an assembling technique that integrates both the original DINOv2 and its own model. Moreover, Fig. 5 shows that our estimated depths are clearer and remain robust across challenging inputs (*e.g.*, severe illumination changes). In contrast, other methods produce unintended noise in extremely dark or bright pixels.

**Semantic segmentation.** Table 3 shows the linear probing results on the Cityscapes (Cordts et al., 2016) and ADE20k (Zhou et al., 2017) datasets. Our method achieves better mIoU than DINOv2 and Condense (Zhang et al., 2024), demonstrating strong transferability to out-of-domain segmentation tasks. Furthermore, our method improves performance not only on day-clear images (as shown in the

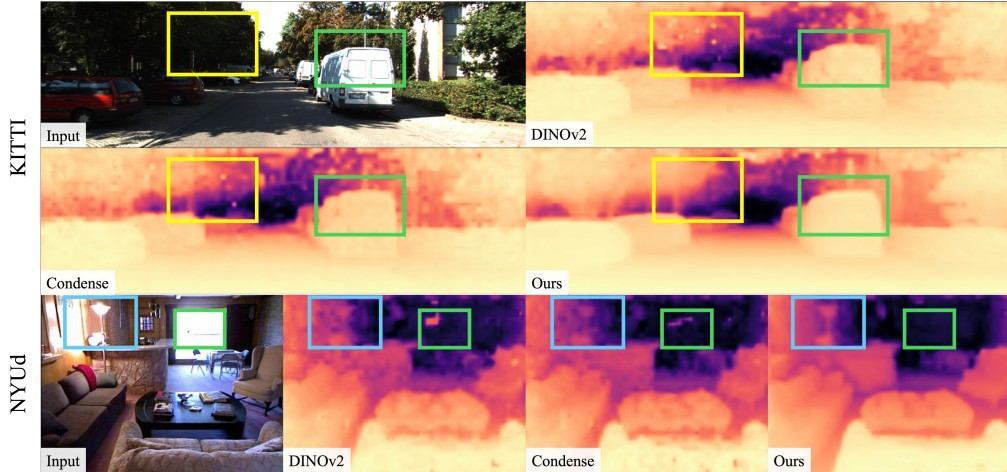

Figure 5: **Qualitative results of out-of-domain depth estimation**. Compared with DINOv2 and Condense (Zhang et al., 2024), our method yields clearer and less noisy depth maps in boxed areas, exhibiting robustness and strong generalization despite being pre-trained only on outdoor.

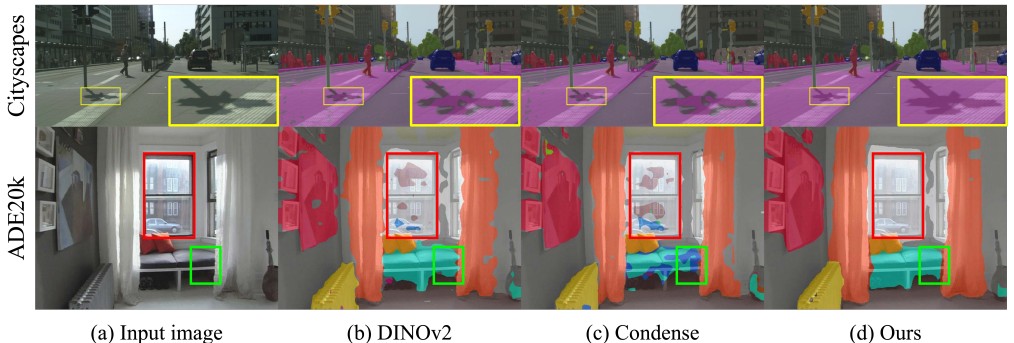

(a) Input image          (b) DINOv2          (c) Condense          (d) Ours

Figure 6: **Qualitative results of out-of-domain semantic segmentation**. Compared with DINOv2 and Condense (Zhang et al., 2024), our method achieves clearer and less noisy segmentation under illumination changes (yellow box) and preserves semantic context for discriminative regions (red/green boxes), demonstrating robustness and strong generalization despite being pre-trained only on outdoor.

in-domain semantic segmentation results in Sec. 4.2) but also in indoor scenarios. This demonstrates that it preserves the 2D encoder's semantic context while making it more discriminative with 3D depth information. The strength of our method is further supported by the qualitative examples in Fig. 6. In the top row, the input image contains a dark shadow on the road. Our method identifies the semantics (i.e., the road) despite the illumination change, whereas other methods fail to capture the semantics and rely on pixel-intensity differences. The bottom row further highlights its ability to capture more discriminative semantics.

**Multi-task learning.** In Table 4, we report the multi-task learning performance in day-rain and night scenes using Video Panoptic Quality (VPQ) (Kim et al., 2020) for video panoptic segmentation and Root Mean Squared Error (RMSE) for depth estimation. The model with our 2D encoder outperforms the model with DINO (Caron et al., 2021) in both metrics. This adaptation to joint multi-task learning further demonstrates robustness of our method under adverse conditions.

## 4.4 ABLATION STUDIES

**Impact of adverse weather data in Stage 1.** To investigate the effect of including adverse conditions in Stage 1, we compare the semantic segmentation performance of the 3D encoder (after Stage 1) under two conditions: day-clear vs. full data (see Table 5). The results indicate that constructing

| Stage 1 data | Eval. data | Seg. (mIoU ↑) |
|---|---|---|
| full day-clear | day-rain | 60.0 **61.1** |
| full day-clear | night | 42.8 **48.0** |

Table 5: **Performance of 3D encoder across Stage 1 data selections**

| Method | Eval. data | Seg. (mIoU ↑) | Depth (RMSE ↓) |
|---|---|---|---|
| DINOv2 + CD | Fog | 43.4 **43.9** | 8.94 **8.16** |
| DINOv2 + CD | Gaussian | 28.4 **32.2** | 10.76 **9.35** |
| DINOv2 + CD | Motion Blur | 44.6 **47.1** | 9.23 **8.69** |

Table 6: **Performance on diverse corrupted images.**

| Method | Seg. (mIoU ↑) nuScenes | Depth (RMSE ↓) nuScenes | KITTI | NYUd |
|---|---|---|---|---|
| DINOv2 | 49.2 | 8.37 | 2.99 | 0.443 |
| DINOv2+ | 49.4 | 8.21 | 3.03 | 0.457 |
| CD | **51.9** | **7.64** | **2.80** | **0.431** |

Table 7: **Comparison of further trained DINOv2 (denoted by DINOv2+) with our method.**

| Method | Stage 1 | Stage 2 | Seg. (mIoU ↑) nuScenes | nuImages | Depth. (RMSE ↓) nuScenes |
|---|---|---|---|---|---|
| DINOv2 | - | - | 49.2 | 70.9 | 8.37 |
| + CD | ✗ | ✓ | 32.2 | 39.2 | 9.53 |
| + CD | ✓ | ✓ | **51.9** | **71.7** | **7.64** |

Table 8: **Performance with and without stage 1.**

Stage 1 with day-clear images yields superior performance on adverse conditions. This supports our assumptions: (i) consistent with **Assumption 1 (Day-clear reliability)**, the 2D encoder pre-trained predominantly on day-clear images provides unreliable features under adverse conditions, which can hinder 3D encoder training; and (ii) consistent with **Assumption 2 (3D-anchored stability)**, even when Stage 1 is trained only on day-clear images, the 3D encoder retains feature resilience under adverse conditions owe to the distributional similarity to day-clear data.

**Ablation on input corruptions.** Beyond adverse-weather settings (night and day-rain), we evaluate robustness to additional image corruptions (*e.g.*, fog, Gaussian noise, and motion blur). Table 6 shows that our method consistently outperforms DINOv2 across all corruptions, indicating robust generalization to diverse degraded inputs.

**Pre-training by CD vs. DINOv2 protocol.** One may question whether the performance improvements reported in Section 4.2 stem primarily from additional training on the nuScenes dataset rather than from the merits of our method. To answer that question, we further train the pre-trained DINOv2 on the nuScenes image data, using the same training protocol of DINOv2. Table 7 shows that the further trained DINOv2 still significantly underperforms compared to our method. This result demonstrates the effectiveness of our method, which is specifically designed to preserve the 2D encoder's original capabilities while enhancing robustness. It also supports the view that self-supervision from 2D images alone is limited in improving robustness to noisy and adverse inputs.

**Effectiveness of Stage 1.** The objective of our two-stage design is to ensure the 2D encoder preserves its original semantic context. To verify the effectiveness of Stage 1, we present a baseline that directly injects the LiDAR information into the 2D image encoder without Stage 1 and only with Stage 2. For this, we train the 3D encoder from scratch on 3D semantic segmentation and then apply Stage 2 without Stage 1. Then, we compare this baseline with our method on in-domain downstream tasks. Table 8 shows that removing Stage 1 results in a significant performance drop. This demonstrates that Stage 1 is crucial to preserve the original semantic context in the collaborative distillation.

## 5 CONCLUSION

In this paper, we present a self-supervised collaborative distillation method to improve 2D image encoders under noisy and adverse conditions, with 3D awareness obtained as a by-product. Our approach demonstrates overall improvements across day/night, in-domain/out-of-domain, and outdoor/indoor scenarios. These improvements are achieved through our carefully designed two-stage paradigm, which fully harnesses image–LiDAR pairs from a single outdoor driving dataset. The 2D image encoder pre-trained on a single dataset successfully transferring across diverse domains demonstrates its potential for generalization ability and adaptability in various vision-based autonomous systems. This success suggests future research directions, such as exploring its generalization ability to new environments and integrating it with different sensor modalities with complementary characteristics.

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

APPENDIX

In this supplementary material, we include additional qualitative results, additional experiments, and experimental details, which could not be included in the main paper because of space limitations.

## A  ADDITIONAL QUALITATIVE RESULTS

### A.1  DOWNSTREAM TASKS QUALITY ON IN-DOMAIN

We compare the depth and segmentation quality of our method with others on in-domain data from the nuScenes dataset, using its day-clear, day-rain, and night validation images. We use the ViT-G/14 model and present linear probing results for depth and segmentation tasks. Our method achieves superior depth and segmentation performance compared to other methods. (See Figs. A and B).

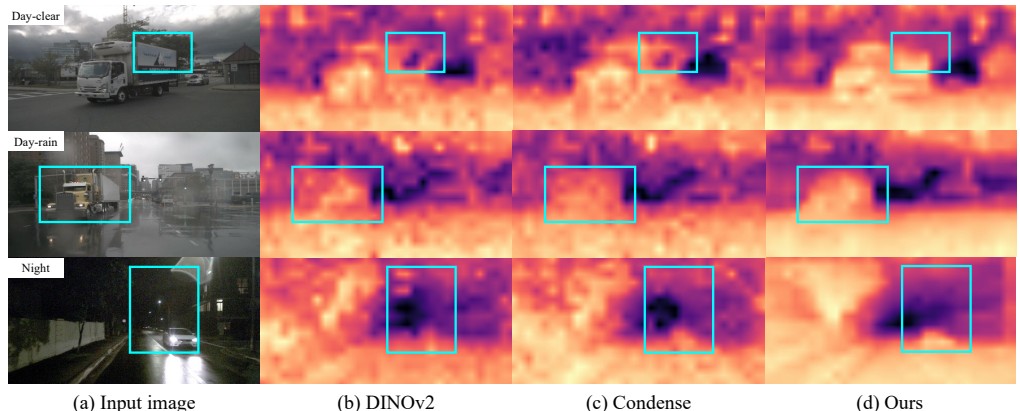

Figure A: **Depth estimation on in-domain**. We present depth estimation quality on the nuScenes dataset for (a) input images, (b) DINOv2, (c) Condense Zhang et al. (2024), and (d) our method. The highlighted regions (cyan box) show that our method produces more accurate and cleaner depth maps than other methods, even under adverse weather conditions.

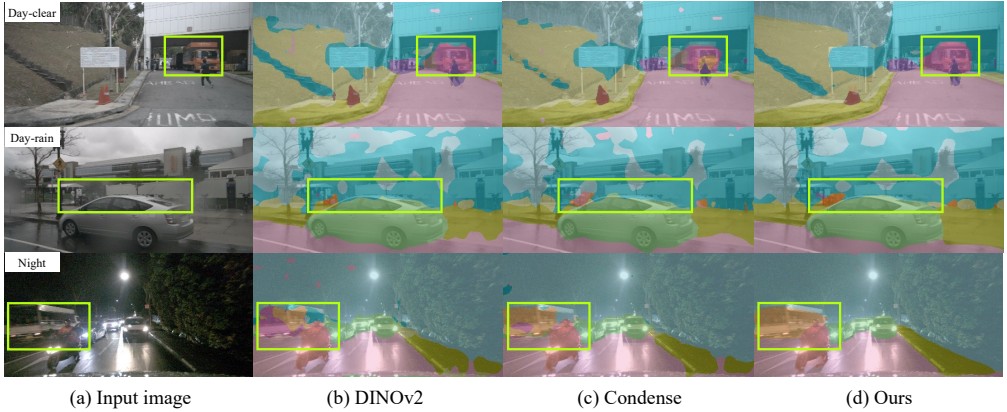

Figure B: **Semantic segmentation on in-domain**. We show semantic segmentation quality on nuScenes dataset for (a) input images, (b) DINOv2, (c) Condense Zhang et al. (2024), and (d) our method. The highlighted regions (green box) show that our method achieves superior segmentation quality than other methods, even under adverse weather conditions.

| Method | Arch. | Seg. (mIoU ↑) | | Depth. (RMSE ↓) |
|---|---|---|---|---|
| | | nuScenes | nuImages | nuScenes |
| MoCov2 (Chen et al., 2020b) | ResNet50 | 38.7 | 55.3 | 9.22 |
| + CD | | **41.4** | **55.8** | **8.19** |
| DINO (Caron et al., 2021) | ResNet50 | 38.4 | 55.8 | 9.01 |
| + CD | | **46.1** | **61.3** | **7.83** |
| DINO (Caron et al., 2021) | ViT-B/8 | 44.7 | 62.4 | 8.93 |
| + CD | | **48.5** | **66.1** | **8.16** |
| IBOT (Zhou et al., 2022) | ViT-L/16 | 47.5 | 65.9 | 8.32 |
| + CD | | **49.3** | **67.7** | **7.79** |
| CLIP (Radford et al., 2021) | ViT-B/16 | 45.8 | 65.8 | 9.05 |
| + CD | | **49.2** | **67.9** | **7.93** |
| DepthAnythingV2 (Yang et al., 2024c) | ViT-B/14 | 53.5 | 75.1 | 7.32 |
| + CD | | **55.1** | **75.8** | **6.86** |
| DINOv3 (Siméoni et al., 2025) | ViT-H+/16 | 64.5 | 83.7 | 5.21 |
| + CD | | **65.4** | **84.2** | **5.20** |

Table A: **Performance of additional methods and architectures.** We report the linear probing results of the other self-supervised methods, architecture, and non-self-supervised method on in-domain datasets. The results show that our method improves performances in all metrics, demonstrating that our method is effectively applied to various methods and architectures.

# B ADDITIONAL EXPERIMENTS

## B.1 ADDITIONAL METHODS AND ARCHITECTURES

To investigate whether our method is compatible with various methods and architectures, we apply our method to various self-supervised learning methods, model architectures, and even non-self-supervised methods. Table A shows that our proposed method outperforms across all metrics when applied to various self-supervised pre-training methods (Chen et al., 2020b; Caron et al., 2021; Zhou et al., 2022; Siméoni et al., 2025), encoder architectures (He et al., 2016; Dosovitskiy, 2020), and even non-self-supervised methods such as CLIP (Radford et al., 2021) and the foundation depth estimation model DepthAnythingV2 (Yang et al., 2024c). DepthAnythingV2 follows an image-encoder + depth-decoder structure, so we use its pre-trained image encoder as the starting point for our method from the official checkpoint. Interestingly, when applied to DepthAnythingV2, our method shows that using a self-supervised pre-trained image encoder such as DINOv2 can even surpass a depth-specialized foundation model. This shows our method's versatility across self-supervised methods, encoder architectures, and even non-self-supervised methods, underscoring its broad and practical applicability.

## B.2 SENSITIVITY STUDY ON THE LiDAR QUALITY

To study how LiDAR sparsity and noise affect our method, we corrupt the LiDAR data using the official code from (Dong et al., 2023a). We apply Gaussian noise and density-decrease corruptions with different severity levels and evaluate our method under each setting. Table B summarizes linear probing results on nuScenes, where severity levels follow (Dong et al., 2023a). We observe that stronger corruption leads to larger performance drops, confirming that LiDAR quality influences the distillation signal. Nevertheless, moderate corruption still preserves most of the gains from our method, and we empirically observe that depth estimation even improves under certain corruption levels.

### B.3    Additional Adverse-Weather Test Set

We train and evaluate the DINOv2 ViT-G model and our method on ACDC (Sakaridis et al., 2021), a real adverse-weather dataset that includes night, rain, fog, and snow. We also evaluate the model trained on Cityscapes directly on the ACDC validation splits without any additional training to clarify its OOD robustness. Table C shows that our approach remains effective on another real-world dataset with diverse adverse conditions.

### B.4    Extention of multiple dataset

We verify the scalability of our method by examining whether it can operate effectively on multiple datasets. First, we train the DINOv2 ViT-L model in Stage 2 using the ScaLR (Puy et al., 2024) checkpoint, which is trained across multiple datasets (nuScenes (Caesar et al., 2020), KITTI (Geiger et al., 2012), Pandar64, and PandarGT (Xiao et al., 2021)). Second, in Stage 1, we exclude adverse-weather-condition data and proceed with the same process. Table D confirms that our method is not limited to nuScenes and can benefit from multi-dataset training. Furthermore, the data-split strategy is also effective in the multi-dataset scenario.

### B.5    Extending Comparison to LiDAR-Based Approaches

We conduct additional experiments, extending the comparison beyond FiT3D and Condense, to ensure a consistent use of training data (*e.g.*, paired LiDAR–image data). We consider BEVDistill (Chen et al., 2022) and DistillBEV (Chen et al., 2022), which meet the following conditions: (1) they use both LiDAR and image data, and (2) their image encoders are trained. Both BEVDistill and DistillBEV train on nuScenes (in-domain) for 3D object detection using LiDAR and images, and their image encoders (ResNet50) are updated during training. We linear probe their officially released image encoder weights on nuScenes semantic segmentation and depth estimation in an in-domain setting. For a fair comparison, we also linear probe ResNet50 pre-trained by DINO and the same backbone further trained by our method. (Note that there are no ResNet50 pre-trained by DINOv2) Table E shows that our method brings larger performance gains on downstream tasks, even compared to methods that use LiDAR data

### B.6    Camera-based 3D object detection

We conduct an additional experiment on a camera-based 3D object detection task to further assess the 3D awareness of the proposed method. The PETR (Liu et al., 2022) decoder remains trainable while the image backbone is replaced with ViT-S pre-trained by DINOv2, FiT3D, or our method and then frozen. Training and evaluation are performed on nuScenes, following standard 3D-detection metrics (mAP and NDS), and all experiments are conducted using the MMDetection3D[2] (Contributors, 2020a) repository. Table F shows that FiT3D performs worse than DINOv2, whereas the proposed method consistently improves over DINOv2 on both metrics. This result strengthens the evidence that the proposed method enhances 3D-aware 2D representations and benefits practical automotive tasks such as camera-based 3D object detection.

### B.7    Training Epochs in Stage 2

We compare the linear probing performances of our method when trained with more epochs in Stage 2. Table G shows that as the number of epochs increases, in-domain performance improves, but out-of-domain performance decreases, indicating a drop in generalization ability. In other words, more epochs lead to overfitting on the pre-training dataset. Since DINOv2 is already trained on abundant data, only one epoch is needed to enhance the learned 2D representations without losing general transferability. This behavior was also observed in FiT3D (Yue et al., 2024).

---

[2]https://github.com/open-mmlab/mmdetection3d

| Method | LiDAR Corruption | Severity | Seg. (mIoU ↑) | Depth. (RMSE ↓) |
|---|---|---|---|---|
| DINOv2 | - | - | 49.2 | 8.37 |
| + CD | ✗ | - | **51.9** | 7.64 |
| + CD | Gaussian Noise | 1 | 51.6 | 7.65 |
| + CD | | 2 | 50.4 | **7.59** |
| + CD | Density Decrease | 1 | **51.9** | 7.66 |
| + CD | | 2 | 51.7 | 7.67 |

Table B: **Performance on diverse LiDAR corruptions with levels.**

| Method | Train. data | Eval. data | Seg. (mIoU ↑) |
|---|---|---|---|
| DINOv2 | ACDC Night | ACDC Night | 52.8 |
| + CD | | | **53.1** |
| DINOv2 | ACDC Rain | ACDC Rain | 60.7 |
| + CD | | | **68.0** |
| DINOv2 | ACDC Fog | ACDC Fog | 66.4 |
| + CD | | | **74.3** |
| DINOv2 | ACDC Snow | ACDC Snow | 64.5 |
| + CD | | | **70.1** |
| DINOv2 | CityScapes | ACDC Night | 46.4 |
| + CD | | | **49.0** |
| DINOv2 | CityScapes | ACDC Rain | 63.1 |
| + CD | | | **66.9** |
| DINOv2 | CityScapes | ACDC Fog | 69.9 |
| + CD | | | **72.1** |
| DINOv2 | CityScapes | ACDC Snow | 62.3 |
| + CD | | | **66.7** |

Table C: **Performance on diverse adverse weather images on ACDC (Sakaridis et al., 2021)**

| Method | Stage 1 Dataset | Stage 2 Dataset | Seg. (mIoU ↑) | Depth. (RMSE ↓) |
|---|---|---|---|---|
| DINOv2 | - | - | 53.6 | 7.97 |
| + CD | Multiple | Multiple | 57.1 | 6.96 |
| + CD | Multiple-clear | Multiple | **57.5** | **6.95** |

Table D: **Extension of multiple datasets.** Multiple denotes the dataset that combines nuScenes (Caesar et al., 2020), KITTI (Geiger et al., 2012), Pandar64, and PandarGT (Xiao et al., 2021). Multiple-clear denotes the Multiple dataset without adverse-weather conditions. Note that the Stage 1's result in the second row is obtained from the ScaLR (Puy et al., 2024) official checkpoint.

| Method | Arch. | Seg. (mIoU ↑) | Depth (RMSE ↓) |
|---|---|---|---|
| DINO (Caron et al., 2021) | ResNet50 | 38.4 | 9.01 |
| + CD | | **46.1** | **7.83** |
| BEVDistill (Chen et al., 2022) | ResNet50 | 36.1 | 9.93 |
| DistillBEV (Wang et al., 2023) | | 44.9 | 9.11 |

Table E: **In-domain comparison with LiDAR-based methods**

### B.8 HYPERPARAMETER SEARCH FOR DINOV2

To investigate whether the further trained model by DINOv2 still shows limited performance with different learning rates and more epochs, we train and validate this model with various hyperparameter settings. The model architecture used is ViT-S/14. First, while further training the image encoder by the DINOv2 method, we fix the epoch to 1 (as in our method's Stage 2) and train with various learning rates. We select the learning rate that achieves peak validation performance across downstream tasks. From $\{5.0 \times 10^{-6}, 1.0 \times 10^{-5}, 2.0 \times 10^{-5}, 5.0 \times 10^{-5}, 1.0 \times 10^{-4}\}$, we select $2.0 \times 10^{-5}$. Second, considering this model with one epoch might be insufficient, we train the model with 25 epochs using the selected learning rate. Table H shows that even with extended epochs, the model does not achieve the overall performance improvement. Consistent with Table G, increasing epochs improves in-domain performance but harms generalization on out-of-domain datasets. These results demonstrate that our method cannot be replaced by a trivial extension even after an exhaustive hyperparameter search for the model. The improved performance of our method stems from our unique design to improve the 2D encoder rather than just additional training on the nuScenes.

### B.9 ADDITIONAL EVALUATION OF COMPETING METHODS

We evaluate the additional generalizability of the other methods FiT3D (Yue et al., 2024) and Condense (Zhang et al., 2024). First, to ensure a fair comparison in model size, we reproduce the FIT3D, which does not use assembling. Table I shows that both FiT3D and Condense perform worse in generalization compared to the DINOv2. However, FiT3D with the assembling method achieves better generalization performance. Improving the generalization ability of image encoders, including assembling methods, represents a significant avenue for future investigation.

### B.10 LINEAR HEAD CONFIGURATIONS IN STAGES 1&2

We investigate how the design of linear heads attached to the 2D and 3D encoders at different stages affects performance. In Stage 1, we do not use a 2D head and only use a 3D head. If a 2D head were used instead of a 3D head to align the dimensions, the randomly initialized 2D head would disrupt

| Method | Decoder | mAP ↑ | NDS ↑ |
|---|---|---|---|
| DINOv2 | | 10.6 | 19.0 |
| + FiT3D | PETR (Liu et al., 2022) | 6.1 | 16.2 |
| + CD | | **11.5** | **19.2** |

Table F: **3D object detection results on nuScenes.**

| Method | Stage 2 epochs | Seg. (mIoU ↑) | Depth. (RMSE ↓) | | |
|---|---|---|---|---|---|
| | | nuScenes | nuScenes | KITTI | NYUd |
| DINOv2 | - | 49.2 | 8.37 | 2.99 | 0.443 |
| + CD | 25 | **52.7** | **7.44** | 3.08 | 0.541 |
| + CD* | 1 | 51.9 | 7.64 | **2.80** | **0.434** |

Table G: **Performance of Collaborative Distillation with different epochs.** We report the linear probing performances of our method with different epochs. The ∗ denotes the version we chose. The results show that our method, with more epochs, achieves high in-domain but low out-of-domain performance with low generalization ability. Our method with one epoch is enough to improve general 2D representations.

| Pre-train Method | Fine-tune Method | Fine-tune Epochs | Seg. (mIoU ↑) | Depth. (RMSE ↓) | | |
|---|---|---|---|---|---|---|
| | | | nuScenes | nuScenes | KITTI | NYUd |
| | - | - | 49.2 | 8.37 | 2.99 | 0.443 |
| | DINOv2 | 25 | 48.7 | 8.10 | 3.16 | 0.533 |
| DINOv2 | DINOv2 | 1 | 49.4 | 8.21 | 3.03 | 0.457 |
| | CD | 25 | **52.7** | **7.44** | 3.08 | 0.541 |
| | CD* | 1 | 51.9 | 7.64 | **2.80** | **0.434** |

Table H: **Performance of DINOv2 with more epochs.** We evaluate the linear probing performance of a DINOv2 with different further training epochs on in-domain (nuScenes) and out-of-domain datasets (KITTI, NYUd). The ∗ denotes the version we chose. The results show that increasing epochs to 25 improves in-domain performance but harms generalization on out-of-domain datasets. The results indicate that a simple extension cannot replace our method, even with a thorough hyperparameter search. Therefore, the improvement of our method comes from our unique design rather than just additional training on the nuScenes

the representations of the 2D image encoder during the early stages of training. Using both a 2D head and a 3D head is not an option, as it would cause a trivial solution (*i.e.*, collapse) due to the nature of the distillation loss we employ. In the same context, in Stage 2, introducing a new head configuration could disrupt the representations during the early stages of training. Therefore, maintaining the same head configuration in stages 1 and 2 ensures stable training and improves performance. Table J shows that using the same head configuration in stages 1 and 2 is effective. These results support the claim that, to preserve the 2D encoder's original capabilities and provide reliable self-supervision, it is crucial to keep the Stage 1 loss and weights unchanged while only switching the gradient direction.

### B.11 JOINT TRAINING WITH ONE STAGE

We investigate a one-stage setup where the 2D and 3D encoders are trained together. With our distillation loss, this joint training easily falls into trivial solutions (*e.g.*., feature collapse). Even when replacing the L2 loss with a contrastive loss and increasing the training epochs, the one-stage setup remains unstable and performs worse than our two-stage setup (See Table K).

## C EXPERIMENTAL DETAILS

### C.1 COMPUTATIONAL OVERHEAD

We report the computational overhead on pre-training stages of proposed method. Table L summarizes the trainable parameters, per GPU memory usage, and GPU-hours for each architecture.

| Method | Arch. | Seg. (mIoU ↑) | | Depth. (RMSE ↓) |
|---|---|---|---|---|
| | | nuScenes | nuImages | nuScenes |
| DINOv2 | | 52.3 | 74.9 | 8.01 |
| + FiT3D[†] (Yue et al., 2024) | ViT-B/14 | 54.6 | 75.7 | 7.60 |
| + FiT3D (Yue et al., 2024) | | 49.7 | 69.3 | 8.12 |
| + CD | | **55.6** | **76.3** | **7.23** |
| DINOv2 | | 55.1 | 77.7 | 7.66 |
| + Condense (Zhang et al., 2024) | ViT-G/14 | 54.7 | 76.7 | 7.66 |
| + CD | | **58.8** | **79.4** | **6.63** |

Table I: **Linear probing performance on nuScenes and nuImages datasets with competitors.** We evaluate the generalization performance of FiT3D and Condense on downstream tasks for nuScenes and nuImages. The result shows that both FiT3D and Condense perform worse than DINOv2. However, FiT3D with the assembling method (†) achieves improved generalization performance thanks to the assembling method.

| Method | 2D head | 3D head | Seg. (mIoU ↑) | Depth. (RMSE ↓) | | |
|---|---|---|---|---|---|---|
| | | | nuScenes | nuScenes | KITTI | NYUd |
| (1) CD | ✓ | ✗ | 45.3 | 7.94 | 3.19 | 0.595 |
| (2) CD* | ✗ | ✓ | **51.9** | **7.64** | **2.80** | **0.434** |

Table J: **Performance of different head configurations.** We report the linear probing performances of the different pretext task head configurations. The $*$ means the version we chose. The results show that (2) configuration is the best, demonstrating that retaining pretext-task head configurations in two stages of our method is important to prevent training disruptions.

## C.2 VISUALIZATION DETAILS

**t-SNE visualization.** We randomly sample 100 images each from the day-clear and night validation sets of the nuScenes (Caesar et al., 2020). [CLS] token features are extracted from the image encoders and visualized using t-SNE. The model architecture used is ViT-S/14. Since the data and feature distributions between day-clear and night are significantly different, we use these two conditions for visualization.

**Feature visualization.** We use the ViT-S/14-reg (Darcet et al., 2023) model for visualization to reduce the influence of positional encoding, which is reflected in the features of the DINOv2-trained ViT model. The colors of features are obtained using principal component analysis (PCA).

## C.3 IN-DOMAIN DOWNSTREAM TASK

We use the AdamW optimizer (Loshchilov, 2017) for all in-domain downstream tasks. Training is done for 10 epochs with a weight decay of $10^{-4}$, and cosine annealing is used as the scheduler. Image augmentation is not applied, and images are resized to $224 \times 448$. All experiments are conducted on a single NVIDIA A100 GPU, except for fine-tuning ViT-G/14, which uses 4 NVIDIA A100 GPUs.

**2D semantic segmentation on nuScenes.** The batch size is 4. The learning rate is $10^{-5}$ for ViT fine-tuning and $5.0 \times 10^{-4}$ for linear probing. ResNet-50 uses a learning rate of $10^{-2}$ for linear probing. Lovasz and cross-entropy loss are used. There are 16 total classes, but when evaluating the night validation set, three classes (bus, trailer, and const vehicle) are excluded as they are not present in the set. Images from all 6 surround cameras in nuScenes are used.

**Depth estimation on nuScenes.** The batch size is 16. The learning rate is $2.0 \times 10^{-5}$ for ViT fine-tuning and $10^{-2}$ for linear probing. ResNet-50 uses a learning rate of $10^{-1}$ for linear probing. We use only MSE loss. The evaluation follows the same split as md4all (Gasperini et al., 2023b), dividing the validation set into day-clear, day-rain, and night. Only the front camera images are used, as in md4all.

| Method | 2-Stages | Epochs | Seg. (mIoU ↑) | Depth. (RMSE ↓) |
|--------|----------|--------|---------------|-----------------|
| DINOv2 | - | - | 49.2 | 8.37 |
| + CD | ✗ | 1 | 38.5 | 7.77 |
| + CD | ✗ | 5 | 30.1 | 7.95 |
| + CD | ✗ | 10 | 25.8 | 8.20 |
| + CD | ✓ | 1 | **51.9** | **7.64** |

Table K: **With and without 2-stages approach.**

| Stage | Arch. | Trainable Params (M) | GPU mem (GB / GPU) | GPU-hours (h) |
|-------|-------|---------------------|--------------------|--------------| 
| Stage1 | ViT-S | 59.7 | 26.4 | 113.6 |
| | ViT-B | 60.0 | 26.6 | 114.4 |
| | ViT-L | 60.2 | 26.8 | 116.4 |
| | ViT-G | 60.6 | 30.8 | 123.1 |
| Stage2 | ViT-S | 22.1 | 15.7 | 4.48 |
| | ViT-B | 86.6 | 19.9 | 4.55 |
| | ViT-L | 304.4 | 31.8 | 4.74 |
| | ViT-G | 1136.5 | 75.2 | 5.17 |

Table L: **Computational overhead of Stage 1 & 2.**

**2D semantic segmentation on nuImages.** The batch size is 24. The learning rate is $2.0 \times 10^{-5}$ for ViT fine-tuning and $5.0 \times 10^{-3}$ for linear probing. ResNet-50 uses a learning rate of $10^{-2}$ for linear probing. Lovasz and cross-entropy loss are used. There are 11 total classes. Day-rain and night data is not evaluated separately due to the small size of the validation set. Images from all six surround cameras in nuImages are used.

C.4    OUT-OF-DOMAIN DOWNSTREAM TASK

**Depth estimation.** We use the Monocular Depth Estimation toolbox[3] (Li, 2022) for out-of-domain depth estimation on KITTI and NYUd. The default hyperparameter configuration from the repository is used. All experiments were conducted on a single NVIDIA A100 GPU.

**2D semantic segmentation.** We use mmsegmentation[4] (Contributors, 2020b) for out-of-domain semantic segmentation on Cityscapes and ADE20k. The default hyperparameter configuration from the repository is used. All experiments are conducted on a single NVIDIA A100 GPU.

**Multi-task learning.** We use Detectron2 (Wu et al., 2019) to implement the depth-aware video panoptic segmentation model (Ji-Yeon et al., 2024). We adopt a small version with one iterative round of Transformer decoder blocks. We train the model using images from Cityscapes (Cordts et al., 2016) and evaluate it on day-rain and night scenarios which are translated by generative method (Parmar et al., 2024). We train the model for 9K iterations with a batch size 16. The learning rate is $5.0 \times 10^{-4}$. We use AdamW optimizer Loshchilov (2017) and polynomial learning rate decay. We use the default setting of the repository for data augmentation and loss balancing. The evaluations are conducted on a single NVIDIA A6000 GPU.

---

[3]https://github.com/zhyever/Monocular-Depth-Estimation-Toolbox
[4]https://github.com/open-mmlab/mmsegmentation

