# OpenReview forum: "LiDAR-Anchored Collaborative Distillation for Robust 2D Representations"
_ICLR.cc/2026/Conference — Submitted to ICLR 2026_

### Official Review · Reviewer_fJFr · 2025-10-27

**Soundness:** 2
**Presentation:** 3
**Contribution:** 2
**Rating:** 4
**Confidence:** 3

**Summary:**

This paper proposes Collaborative Distillation to improve the adaption ability of 2D image encoder with 3D awareness by leveraging 3D LiDAR as self-supervision. It improves the robustness of 2D encoders to noisy and adverse weather conditions. Experiments show that this approach outperforms competing methods in various downstream tasks, highlighting the capability in real-world scenarios.

**Strengths:**

1.	This paper is well-written and easy-to-follow. Figures are clear for the visualization of design and effectiveness.
2.	It is reasonable to utilize LiDAR distillation to improve the robustness of the domain adaptation ability of the 2D vision encoder.
3.	Experiments on 2D semantic segmentation and monocular depth estimation are extensive to show the effectiveness of collaborative distillation.

**Weaknesses:**

1.	As adverse conditions are utilized for self-supervised training, the setting in your paper seems to belong to the field of (unsupervised) domain adaptation, which has been investigated previously, like [1][2][3]. Or, the distinction of these settings should be illustrated clearly.
2.	In experiments, only the CD approach is utilized for comparison based on the visual foundation models: ViT pretrained on DINOv2. Not aiming for domain adaptation, a lot of classic self-supervised representation learning approaches have been proposed. These approaches should also be compared to verify your specific ability for the robustness of domain adaptation.
3.	Experiments are conducted based on DINOv2. More baselines and pre-trained foundation models can be applied to verify the generalization ability of your design.
4.	The correspondence between 2D and 3D and Distillation from LiDAR to image has been investigated in previous perception works in autonomous driving, which is not new for the representation learning approach.

[1] Feng, Zeyu, Chang Xu, and Dacheng Tao. "Self-supervised representation learning from multi-domain data." Proceedings of the IEEE/CVF International Conference on Computer Vision. 2019.
[2] Achituve, Idan, Haggai Maron, and Gal Chechik. "Self-supervised learning for domain adaptation on point clouds." Proceedings of the IEEE/CVF winter conference on applications of computer vision. 2021.
[3] Xu, Jiaolong, Liang Xiao, and Antonio M. López. "Self-supervised domain adaptation for computer vision tasks." IEEE Access 7 (2019): 156694-156706.

**Questions:**

See Above

---

> ### Author Response · Authors · 2025-11-24
> **Response to Reviewer fJFr's Review**
>
> We thank the reviewer fJFr for the thoughtful strengths highlighted in the review, including the clarity of our writing, the motivation for using LiDAR distillation, and the comprehensive experimental validation. We address the reviewer’s concerns and questions below.
>
> ---
>
> ## **W1: Distinction of domain adaptation settings and ours**
>
> The cited works 1,2,3 address domain adaptation, but our setting and objective differ in several fundamental ways.
>
> These cited works operate either on 2D-only inputs 1,3 or 3D-only inputs 2, and their goal is to align a source domain to a target domain for a specific downstream task.
>
> In contrast, our work focuses on improving a self-supervised 2D foundation model (e.g., DINOv2) by leveraging a calibrated 3D modality during pre-training.
>
> Our input setup (paired camera–LiDAR) and our objective (stronger 2D representations under adverse-weather noise) are therefore different.
>
> Most importantly, our method is not a domain-adaptation approach.
>
> We do not learn a mapping between a source and target domain.
>
> Instead, we aim for domain generalization by injecting useful LiDAR properties into the 2D encoder without using any labels.
>
> Our two-stage collaborative distillation (2D ↔ 3D) is fundamentally different from the standard DA pipeline.
>
> For clarity:
>
> - 1 learns self-supervised 2D features for classification-based DA on ImageNet-scale backbones; it does not address multi-sensor settings or foundation models.
> - 2 performs self-supervision on point clouds only; both inputs and tasks differ from our 2D-representation setting.
> - 3 also focuses on 2D DA with auxiliary pretext tasks (e.g., rotation) rather than multi-modal representation improvement.
>
> In summary, our setting, sensory inputs, and objective—enhancing a pretrained 2D foundation model using LiDAR as a stable anchor—are fundamentally different from classical domain adaptation.
>
> Our work aims at domain generalization, not domain adaptation, and uses a collaborative 2D–3D distillation scheme that prior DA-based works do not explore.
>
> ---
>
> ## **W2: Comparison with self-supervised representation learning and CD approach**
>
> We compare our method with a strong self-supervised baseline by strictly following the original DINOv2 training protocol.
>
> We further train the pre-trained DINOv2 on nuScenes with the same settings and denote it as DINOv2+ in the main paper.
>
> As shown in Section 4.4 (“Pre-training by CD vs. DINOv2 protocol.”) and Table 7, this experiment directly tests whether our performance gains come simply from additional training on nuScenes.
>
> The results clearly show that this is not the case: even with additional training, DINOv2+ achieves significantly lower performance than our method.
>
> ---
>
> ## **W3: More baselines and pre-trained foundation models**
>
> We conduct these experiments in Appendix Section B.1 (“Additional Methods and Architectures”), and the results are presented in Table A.
>
> Our method consistently improves performance across various self-supervised methods, encoder architectures, and even a non-self-supervised methods such as CLIP and the foundation depth estimation model Depth Anything v2.
>
> This demonstrates that our design generalizes well and is broadly applicable.
>
> ---
>
> ## **W4: The Distillation from LiDAR to image is not new**
>
>
> Based on the works cited in our related-work section on “Image-to-LiDAR distillation” and “improving 3D awareness of 2D representations,” prior studies are mostly designed for specific downstream tasks (e.g., detection or depth prediction) or do not attempt to improve the robustness of a self-supervised pre-trained image encoder.
>
> Our key contribution is a two-stage training strategy that transfers useful LiDAR properties into a 2D encoder while preserving its original capabilities, which has not been addressed in previous cross-modal distillation research.
>
> Stage 1 (pre-alignment) and Stage 2 (3D-anchored self-supervision) play complementary roles and explicitly target robustness under noisy and adverse-weather conditions—an aspect that prior approaches do not consider.
>
> Our framework therefore provides a novel formulation of cross-modal learning for representation learning, where a calibrated 3D modality serves as a stable anchor to build more robust 2D features.

---

### Official Review · Reviewer_KQwi · 2025-10-31

**Soundness:** 3
**Presentation:** 3
**Contribution:** 2
**Rating:** 6
**Confidence:** 4

**Summary:**

This manuscript introduces LiDAR-Anchored Collaborative Distillation (CD), a two-stage self-supervised framework designed to improve the robustness of 2D image encoders under adverse weather and noisy conditions. The key idea is to leverage LiDAR representations as stable self-supervision anchors for 2D encoders.

- Stage 1 (Pre-alignment): Aligns LiDAR features with 2D clear-weather features to create a reliable 3D anchor space.
- Stage 2 (3D-anchored supervision): Uses these aligned 3D features to pull degraded 2D features (e.g., night, rain, noise) toward the clean 2D feature space.

The manuscript claims consistent improvements over DINOv2 baselines across multiple downstream tasks (semantic segmentation, depth estimation, and video panoptic segmentation), and generalizes well to both indoor and outdoor out-of-domain datasets.

**Strengths:**

(+) The manuscript addresses an important and realistic gap: the lack of robustness of vision foundation models (e.g., DINOv2) under adverse conditions. The motivation for using LiDAR as a more stable and weather-invariant modality for self-supervision is clearly justified and timely.

(+) Extensive experiments on nuScenes, nuImages, and some other benchmarks (KITTI, NYUd, ADE20k, Cityscapes) confirm that CD consistently improves robustness and generalization. The use of both few-shot and full-label setups adds credibility to the findings.

(+) The manuscript provides several clear figures showing the feature distribution shift toward stable regions. Ablation studies (Stage 1 effect, corruption robustness, etc.) are thorough and support the methodological choices.

**Weaknesses:**

(-) While the framework is well-motivated and cleanly executed, it primarily combines known techniques — cross-modal distillation, bidirectional matching, and stop-gradient supervision — into a new use case. The innovation lies more in formulation and application (LiDAR as a teacher for robustness) than in new architecture or loss design.

(-) Lack of discussions with several closely related works. There is a line of 2D-3D cross-modal knowledge transfer work, which, in this manuscript, was omitted for discussions and/or comparisons. Notable works include xMUDA (CVPR'20), xMoSSDA (TPAMI'22), CLIP2Scene (CVPR'23), SuperFlow (ECCV'24), ScaLR (CVPR'24), and LiMoE (CVPR'25). The authors are suggested to discuss these closely related works to ensure the claims made are grounded.

(-) The approach assumes accurate calibration between LiDAR and cameras, which may not always hold in real-world deployment. Discussion of calibration noise, sparse LiDAR data, or generalization to other modalities (e.g., radar) would strengthen applicability claims.

(-) The manuscript shows consistent performance gains but does not discuss computational overhead, convergence stability, or potential degradation when LiDAR data is unavailable during pretraining.

**Questions:**

In addition to the weaknesses mentioned in the above section, the authors are suggested to clarify on the following minor questions:

(-) Carify whether the “bi-directional distillation” uses shared or separate projection heads and if the stop-gradient operation is symmetric or alternated during training.

(-) Include a small analysis on training stability or feature collapse prevention under the two-stage setup.

(-) Discuss failure cases more concretely (e.g., scenarios where LiDAR sparsity or missing points weaken supervision).

(-) In Fig. 1 and 2, consider explicitly labeling “Stage 1” and “Stage 2” arrows for readability.

(-) In Section 4.3, briefly explain why improvements are stronger on indoor OOD datasets despite pretraining on outdoor-only data.

---

> ### Author Response · Authors · 2025-11-24
> **Response to Reviewer KQwi's Review**
>
> We thank the reviewer KQwi for the thoughtful strengths highlighted in the review, including the clear motivation of addressing robustness gaps in vision foundation models, the extensive experimental validation across multiple benchmarks, and the clarity of the figures and ablation studies. We address the reviewer’s concerns and questions below.
>
> ---
>
> ## **W1: Novelty beyond simply combining existing techniques**
>
> Thank you for the positive feedback regarding the clarity and motivation of our framework.
>
> While our method uses known building blocks (e.g., distillation, matching, stop-gradient), our contribution is not a simple combination of existing techniques.
>
> Our two-stage training strategy is the key novelty.
>
> - Stage 1 (Pre-alignment) and Stage 2 (3D-anchored self-supervision) are designed to play complementary roles in transferring LiDAR properties into the 2D encoder while preserving its original capabilities.
>
> - This staged design has not been explored in prior cross-modal distillation works, which typically use a single-stage pipeline and do not address robustness degradation of self-supervised 2D encoders under adverse weather.
>
> - Our formulation uses the calibrated 3D modality as a stable anchor to maintain semantic consistency, offering a new perspective for representation learning rather than a simple combination of existing techniques.
>
> We believe this provides meaningful insights for robust perception.
>
> Moreover, we show that the method works well across different architectures with only a simple projection head and loss, suggesting promising flexibility and extensibility.
>
> ---
>
> ## **W2: Discussions with several closely related works**
>
> We thank the reviewer for pointing out these closely related works.
>
> We clarify that although these methods also study 2D–3D knowledge transfer, their settings, goals, and modalities differ from ours.
>
> ### **xMUDA / xMoSSDA**
>
> These methods target unsupervised domain adaptation for 3D semantic segmentation using multimodal inputs (image + LiDAR).
>
> Their objective is adapting from one domain to another (e.g., day→night), not improving a 2D self-supervised encoder or robustness of 2D representations.
>
> ### **CLIP2Scene**
>
> Transfers CLIP knowledge from 2D→3D to improve point-cloud models.
>
> Our direction is different: we transfer 3D→2D and preserve the 2D encoder’s capabilities.
>
> ### **SuperFlow**
>
> A temporal extension of SLidR / PPKT-style Image-to-LiDAR pre-training, but still focused on enhancing 3D perception models, not improving 2D foundation models.
>
> ### **LiMoE**
>
> A 3D-centric MoE framework integrating multiple LiDAR representations.
>
> Stage 1 includes image-to-LiDAR pre-training, but does not perform the reverse or aim at improving 2D feature robustness.
>
> ### **Overall**
>
> These works mostly aim at 3D perception or domain adaptation, and typically perform single-stage distillation.
>
> In contrast, our method introduces a two-stage collaborative distillation (2D↔3D) specifically designed to enhance a pre-trained 2D foundation model under noisy and adverse-weather conditions—an aspect not addressed in prior works.
>
> We will add these discussions to the revised related-work section.
>
> ---
>
> ## **W3. Discussion about LiDAR failure cases**
>
> Our method requires a paired and well-calibrated camera–LiDAR data during pre-training.
>
> If such paired data are not available, the method cannot be applied.
>
> In addition, LiDAR in real scenarios can be noisy or significantly sparse, which may weaken the supervision signal.
>
> To examine these failure cases, we corrupt the LiDAR data using the official code from [A1].
>
> We apply Gaussian noise and density-decrease corruptions with different severity levels, and then apply our method on these corrupted LiDAR data.
>
> The table below reports linear probing results on nuScenes, where the severity numbers follow the levels defined in [A1].
>
> | Method  | LiDAR Corruption   | Severity | Seg. (mIoU ↑) |
> |---------|---------------------|----------|---------------|
> | **DINOv2** | -                 | -        | 49.2          |
> | + CD    | ✗                 | -        | **51.9**      |
> | + CD    | Gaussian Noise     | 1        | 51.6          |
> | + CD    | Gaussian Noise     | 2        | 50.4          |
> | + CD    | Density Decrease   | 1        | 51.9          |
> | + CD    | Density Decrease   | 2        | 51.7          |
>
> The results show performance degradation when LiDAR is corrupted, and stronger corruption leads to further drops.
>
> As a future direction, improving the reliability and accuracy of LiDAR itself could further enhance the overall robustness of our framework.
>
> [A1] Dong, Yinpeng, et al. "Benchmarking robustness of 3d object detection to common corruptions." Proceedings of the IEEE/CVF Conference on Computer Vision and Pattern Recognition. 2023.

---

> ### Author Response · Authors · 2025-11-24
>
> ## **W4. Generalization to other modalities**
>
> In this work, we focus on LiDAR to investigate whether its two properties—3D awareness and robustness—can be effectively distilled into an image encoder.
>
>
> We find that pre-training on a single LiDAR-equipped dataset improves performance in various LiDAR-free domains, suggesting that practical availability of LiDAR is less of a concern.
>
>
> Since our framework is not restricted to LiDAR and can operate with any calibrated 3D modality (e.g., radar), we believe similar benefits could be achieved as long as the alternative modality provides complementary information.
>
> ---
>
> ## **W5. Computational overhead**
>
> We report the computational overhead as requested.
>
> As described in Section 4.1, all pre-training is done with 4× NVIDIA A100 GPUs.
>
> Stage 1 trains the 3D encoder for 49 epochs with a batch size of 32, and Stage 2 trains the 2D encoder for 1 epoch with a batch size of 8.
>
> The table below summarizes the trainable parameters, per GPU memory usage, and GPU-hours for each architecture.
>
> | Stage  | Arch. | Trainable Params (M) | GPU mem (GB / GPU) | GPU-hours (h) |
> |--------|--------|-----------------------|---------------------|---------------|
> | **Stage1** | ViT-S | 59.7  | 26.4 | 113.6 |
> |        | ViT-B | 60.0  | 26.6 | 114.4 |
> |        | ViT-L | 60.2  | 26.8 | 116.4 |
> |        | ViT-G | 60.6  | 30.8 | 123.1 |
> | **Stage2** | ViT-S | 22.1  | 15.7 | 4.48 |
> |        | ViT-B | 86.6  | 19.9 | 4.55 |
> |        | ViT-L | 304.4 | 31.8 | 4.74 |
> |        | ViT-G | 1136.5 | 75.2 | 5.17 |
>
>
> ---
>
> ## **Q1. Clarification of the bi-directional distillation**
>
> The bi-directional distillation uses the same projection head in both Stage 1 and Stage 2; the projection head is shared between the two stages.
>
> The stop-gradient operation is applied in an alternating manner:
>
> - In Stage 1, the stop-gradient is applied to the 2D encoder.
> - In Stage 2, it is applied to the 3D encoder and the projection head.
>
> ---
>
> ## **Q2. Training stability**
>
> In our two-stage setup, each stage uses a fully frozen teacher network.
>
> Because the teacher is not trainable, the student can learn in a stable way and converges without feature collapse.
>
> However, if the linear head used to match feature dimensions is configured differently in Stage 1 and Stage 2, the training can become unstable.
>
> In this case, the linear head in Stage 2 is not pre-aligned in Stage 1, which leads to lower performance.
>
> This analysis is provided in Appendix B.10 (“Linear Head Configurations in Stage 1 & 2”) and Table J.
>
> We also test a one-stage setup where the 2D and 3D encoders are trained together.
>
> With our distillation loss, this joint training easily falls into trivial solutions (e.g., feature collapse).
>
> Even when replacing the L2 loss with a contrastive loss and increasing the training epochs, the one-stage setup remains unstable and performs worse than our two-stage setup (see the table below).
>
> | Method  | 2-Stages | Epochs | Seg. (mIoU ↑) |
> |---------|----------|--------|----------------|
> | **DINOv2** | -        | -      | 49.2           |
> | + CD    | ❌       | 1      | 38.5           |
> | + CD    | ❌       | 5      | 30.1           |
> | + CD    | ❌       | 10     | 25.8           |
> | + CD    | ✅       | 1      | **51.9**       |
>
>
> ---
>
> ## **Q3. Explanation of improvements on indoor OOD datasets**
>
> Although our pretraining uses outdoor data only, our method preserves the original generalization ability of DINOv2 while adding 3D structural information from LiDAR.
>
> This 3D structure helps improve OOD semantic understanding and depth estimation, which we suspect is a key reason for the stronger gains observed on indoor OOD datasets.

---

### Official Review · Reviewer_hsWC · 2025-11-01

**Soundness:** 4
**Presentation:** 2
**Contribution:** 2
**Rating:** 4
**Confidence:** 5

**Summary:**

This paper introduces a self-supervised learning pipeline, called Collaborative Distillation, which leverages the ToF sensor's low sensitivity to the adverse weather conditions for the 2D RGB image representation.
The strategy is two-staged. Align the 3D ToF sensor's pointcloud encoder to the pre-trained 2D RGB representation in the clear image, and distill the 3D ToF sensor's pointcloud encoder to the 2D RGB representation of the scenes with adverse conditions. The self-supervised training strategy is utilized for the various semantic tasks (depth estimation, segmentation).

**Strengths:**

The idea is simple and straightforward to understand.
The figure is easy to understand, and the paper is easy to read.,
The work leverages complementary sensor properties, and this is considered to be a viable way to solve the nuisance variables.

**Weaknesses:**

The selection of the Dinov2 is questionable.
The evaluation dataset is limited to the nuScenes dataset.
The t-SNE plot (Fig. 3) is not convincing.
The Waymo dataset is considered another set to test out, which has adverse weather conditions as well.

**Questions:**

1. Usage of the DINOv2.
Why DINOv2?: DINOv2 is supposed to learn the invariant feature under the photometric/geometric perturbations, which may occur in adverse weather conditions as well. Therefore, the linear probing of the DINOv2 feature would not be adequate (given that DepthAnything also trains the DINOv2 features). Another point is that the training data of DINOv2. DINOv2 is trained on object-centric images, not the outdoor driving scenes. Therefore, it is obvious that DINOv2 cannot perform better than DINOv2+ and Ours. From the reviewer's perspective, this undermines the proposed method's significance. Could the authors clarify?

2. What does the t-SNE plot (Fig. 3) imply?
The left two plots (Dino, Dinov2) divide the adverse weather data, and the right plot shows the self-supervised model dividing the adverse weather data into two chunks, and the right does not seem to be better, given that the t-SNE plot is a dimensionality reduction method to plot the high-dimensional data space into just a two-dimensional space. What is the purpose of the Fig. 3?

3. Monocular Depth Estimation model trained with the proposed method?
Connected to question 1, the reviewer was wondering if the proposed self-supervised method can generalize to the MDE models?

4. The sensitivity study on the LiDAR quality
The proposed method leverages the complementary characteristics of the camera and LiDAR, where the camera is dense but sensitive to the domain shift [1], and the LIDAR is robust to the nuisance variables but sparse. This implies that stage 2 will be affected by the LiDAR's sparsity and noise level.
Based on the question, the reviewer hopes to see the sensitivity study on the sparsity/noise of the LiDAR data.

[1] Park et al., "Test-Time Adaptation for Depth Completion," CVPR 2024

---

> ### Author Response · Authors · 2025-11-24
> **Response to Reviewer hsWC's Review**
>
> We thank the reviewer hsWC for the thoughtful strengths highlighted in the review, including the simplicity and clarity of the core idea, the easy-to-follow figures and writing, and the recognition that leveraging complementary sensor properties is a viable way to address nuisance factors in visual perception. We address the reviewer’s concerns and questions below.
>
> ---
>
> ## **Q1: Why DINOv2 and does it undermine contribution?**
>
> We use DINOv2 because it is a widely adopted self-supervised foundation image encoder in current vision research.
>
> It is not clearly validated that DINOv2 is designed to learn features invariant to photometric or geometric perturbations.
>
> Even if such invariance exists, our consistent improvements show that applying our method enables the model to extract features more robust to noisy and adverse-weather conditions.
>
> It is expected that DINOv2+ and our method outperform DINOv2 in in-domain settings, since they are additionally trained.
>
> However, the key observation is that in the out-of-domain setting (Table 7), DINOv2+ does not outperform DINOv2, even though it was further trained on nuScenes.
>
> In contrast, our method preserves the original capabilities of DINOv2 while learning more robust and 3D-enhanced features, which allows it to outperform both DINOv2 and DINOv2+ under OOD conditions.
>
> These results indicate that our method provides generalization benefits beyond what DINOv2 or simple additional training can achieve.
>
> ---
>
> ## **Q2: Implication of t-SNE plot**
>
> In DINOv2, the day-clear and adverse-weather features appear almost fully separated in the representation space, and this separation remains even after further training in DINOv2+.
>
> In contrast, our method produces more overlapped feature distributions.
>
> This indicates that our model learns features that are more invariant across day-clear and adverse-weather inputs.
>
> While t-SNE is indeed a 2D projection, the qualitative trend is consistent: our method reduces the discrepancy between the two distributions, whereas DINOv2 and DINOv2+ keep them far apart.
>
> Similar to Fig. 1 in the main paper, Fig. 3 provides qualitative evidence that our approach learns the intended invariant representations across different weather conditions.
>
> ---
>
> ## **Q3: Generalization to the MDE models**
>
> As requested, we validate this by applying our approach to DepthAnythingv2 [A1], a widely used MDE foundation model.
>
> DepthAnythingv2 follows an image-encoder + depth-decoder structure, so we use its pre-trained image encoder as the starting point for our method from the official checkpoint.
>
> For a fair comparison, both DINOv2 and DepthAnythingv2 use a ViT-B encoder and a linear depth head.
>
> As shown in the table below, our method further improves the performance of DepthAnythingv2 across both segmentation and depth estimation metrics.
>
> Interestingly, the results indicate that applying our method to a self-supervised pre-trained image encoder such as DINOv2 can surpass even a depth-specialized foundation model in depth estimation.
>
> | Method              | Arch.      | Seg. (mIoU ↑)         |               | Depth. (RMSE ↓) |
> |---------------------|------------|------------------------|---------------|------------------|
> |                     |            | **nuScenes**           | **nuImages**  | **nuScenes**     |
> | DINOv2     | ViT-B/14   | 52.3                   | 74.9          | 8.01             |
> | + CD                | ViT-B/14   | **55.5**               | **76.4**      | **7.18**         |
> | DepthAnythingV2     | ViT-B/14   | 53.5                   | 75.1          | 7.32             |
> | + CD                | ViT-B/14   | **55.1**               | **75.8**      | **6.86**         |
>
> [A1] Yang, Lihe, et al. "Depth anything v2." Advances in Neural Information Processing Systems 37 (2024)

---

> ### Author Response · Authors · 2025-11-24
>
> ## **Q4: Sensitivity study on the LiDAR quality**
>
> To study how LiDAR sparsity and noise affect our method, we corrupt the LiDAR data using the official code from [A2].
>
> We apply Gaussian noise and density-decrease corruptions with different severity levels and evaluate our method under each setting.
>
> The table below summarizes linear probing results on nuScenes, where severity levels follow [A2].
>
> | Method  | LiDAR Corruption   | Severity | Seg. (mIoU ↑) |
> |---------|---------------------|----------|---------------|
> | **DINOv2** | -                 | -        | 49.2          |
> | + CD    | ✗                 | -        | **51.9**      |
> | + CD    | Gaussian Noise     | 1        | 51.6          |
> | + CD    | Gaussian Noise     | 2        | 50.4          |
> | + CD    | Density Decrease   | 1        | 51.9          |
> | + CD    | Density Decrease   | 2        | 51.7          |
>
> We observe that stronger corruption leads to larger performance drops, confirming that LiDAR quality influences the distillation signal.
>
> Nevertheless, moderate corruption still preserves most of the gains from our method.
>
> [A2] Dong, Yinpeng, et al. "Benchmarking robustness of 3d object detection to common corruptions." Proceedings of the IEEE/CVF Conference on Computer Vision and Pattern Recognition. 2023.
>
> ---
>
> ## **W1: Another test set**
>
> We train and evaluate our method on ACDC [A3], a real adverse-weather dataset that includes night, rain, fog, and snow.
> We also evaluate the model trained on Cityscapes directly on the ACDC validation splits without any additional training to clarify its OOD robustness.
>
> The results below show that our approach remains effective on another real-world dataset with diverse adverse conditions.
>
> | Method   | Train. data | Eval. data | Seg. (mIoU ↑) |
> |----------|-------------|------------|----------------|
> | **DINOv2** | ACDC Night | ACDC Night | 52.8 |
> | + CD     | ACDC Night | ACDC Night | **53.1** |
> | **DINOv2** | ACDC Rain  | ACDC Rain  | 60.7 |
> | + CD     | ACDC Rain  | ACDC Rain  | **68.0** |
> | **DINOv2** | ACDC Fog   | ACDC Fog   | 66.4 |
> | + CD     | ACDC Fog   | ACDC Fog   | **74.3** |
> | **DINOv2** | ACDC Snow  | ACDC Snow  | 64.5 |
> | + CD     | ACDC Snow  | ACDC Snow  | **70.1** |
> | **DINOv2** | CityScapes | ACDC Night | 46.4 |
> | + CD     | CityScapes | ACDC Night | **49.0** |
> | **DINOv2** | CityScapes | ACDC Rain  | 63.1 |
> | + CD     | CityScapes | ACDC Rain  | **66.9** |
> | **DINOv2** | CityScapes | ACDC Fog   | 69.9 |
> | + CD     | CityScapes | ACDC Fog   | **72.1** |
> | **DINOv2** | CityScapes | ACDC Snow  | 62.3 |
> | + CD     | CityScapes | ACDC Snow  | **66.7** |
>
> [A3] Sakaridis et al., ACDC: The adverse conditions dataset with correspondences for semantic driving scene understanding, ICCV 2021

---

### Official Review · Reviewer_jfki · 2025-11-01

**Soundness:** 2
**Presentation:** 3
**Contribution:** 2
**Rating:** 4
**Confidence:** 4

**Summary:**

This paper proposes Collaborative Distillation (CD), a self-supervised method to improve the robustness of pretrained 2D image encoders to adverse weather (e.g., rain, low-light, night, corruptions) thanks to guidance from Lidar. The authors argue that 2D models fail because they are pretrained mostly on clear-day images. CD leverages robust 3D Lidar information in a two-stage process. In stage 1, a 3D encoder is aligned to a frozen 2D encoder's clear-day features, creating a "reliable anchor". In stage 2, the 2D encoder is finetuned on all available data (including clear and adverse) by distillation from the frozen pretrained 3D encoder (anchor). This process "pulls" the 2D features from adverse conditions toward the stable, clear-day feature distribution.

The CD-enhanced 2D encoder is evaluated on multiple downstream tasks, semantic segmentation and depth estimation, demonstrating improved robustness in-domain (on nuScenes) and strong out-of-domain generalization (on KITTI, NYUd, Cityscapes, ADE20K).

**Strengths:**

**Significance**
- This work deals with an important and highly practical problem for real-world applications like autonomous driving: the reliability of perception models under challenging environmental conditions.
- This is definitely an area that is worth studying further


**Clarity**
- I find the paper well written with multiple visualizations, qualitative visualizations, and detailed implementation information.
- The paper is in general easy to follow and the reasoning of the authors is clear.


**Quality**
- The CD approach is evaluated under several settings, in-distribution and out-of-distribution, on multiple tasks and datasets and distribution shifts
- There are numerous other experiments in the appendix with different pretrained 2D encoders and data splits
- CD shows consistent performance boosts across multiple downstream settings, different tasks and datasets

**Weaknesses:**

**Not completely valid assumptions**
- The authors argue that 3D Lidar representations are highly robust to adverse weather conditions compared to 2D image representations. While this is true for night time conditions, Lidar is quite brittle under rain, fog and snow [a], [b], [c].
- Lidar does shine however in precise 3D information and geometry which might be in fact here one of the main contributing factors to performance boosts.
- The authors assume that the LVD-142M dataset upon which DINOv2 was pretrained, is predominantly composed of clear daytime images. The dataset is private and we cannot really know it composition. It could be clear images, but also night, indoor, outdoor, etc., but we can only guess.

**Limited scope of experiments**
- The authors argue the practical side of this approach and its adaptability to real-world conditions. However, from an autonomous driving standpoint (the type of data that CD needs, paired image and lidar data, is currently found mostly in this type of application), the experimental setup is not convincing.
- The bulk of the in-domain experiments are conducted on nuscenes 2D semantic segmentation. Here the annotated point clouds are projected in the image space and used as ground truth labels. However the point cloud is actually sparse on nuscenes (32-beams lidar) and many of the pixels are not labelled for both training and testing, making it quite an imperfect setting. In fact this protocol is actually established by the authors and not really used in the community.
- Relevant experiments on automotive datasets like nuscenes would be BEV 3D object detection or BEV semantic segmentation, as done in several of the related works in this area, such as the ones mentioned in related works section "Improving 3D awareness of 2D representation"

- To go further OOD the authors leverage the synthetic images from Cityscape-DVPS converted to rainy and night with Stable Diffusion. However there are specific Cityscapes related datasets (for semantic segmentation at least) with adverse weather conditions (night, rain, fog, snow) ACDC [f], multiple corruptions Cityscapes-C [g], with aggregated distributions shifts BRAVO challenge [h]. They could be used directly as they are just additional test sets for Cityscapes-trained models.

**Limited scope of baselines**
- In-domain experiments use just DINOv2 as baseline. However the authors mention in related work different methods that improve 3D awareness of 2D representations: UniPAD, BEVDistill, DistillBEV etc., but also OccFeat [i]  (not cited, but related) that distills DINOv2 features projected on point clouds

- The depth estimation experiments are lacking widely-used baselines such as DepthAnythingV2 (it produces relative depth but could be rescaled or fine-tuned)[d] or DepthPro (producing absolute depths)[e]

- The comparisons with other methods is mostly done on OOD settings but even there it's not quite apple-to-apples as the considered methods (FiT3D, Condense) do not use Lidar supervision

**Contributions**
- The stage 1 of CD is presented as a contribution of the authors, however the setup is identical to the one from ScaLR (Puy et al., 2024), excepting the fact that CD trains only on daytime data
- The idea of distilling a Lidar encoder in image encoders has been explored thoroughly in the literature (many of the methods are in fact cited by the authors in the related work) and the approach itself is not really novel. What is novel is the use of self-supervised Lidar encoders and the splitting of the data across stages (day images in stage 1, full data in stage 2).
- For the latter I'm not fully convinced about the claim as it might be specific to nuscenes. It would be informative to test the distillation on other datasets, as done in ScaLR (Puy et al.), where the authors run image to lidar distillation on KITTI, Pandar 64, Pandar GT, etc.
- Besides I'm wondering what if in stage 2, CD uses directly the ScaLR checkpoint distilled trained on several datasets and lidars? The authors show better generalization in this way.


**Missing related works**
- OccFeat [i] is pre-trained by distilling DINO features projected on point clouds, towards improving multi-camera BEV perceptions
- Image-lidar co-teaching has been proposed also for unsupervised domain adaptation with results on both 3D and 2D semantic segmentation [j]


**Minor Misc.**
- Typos:
    + L051: faces
    + in C3, nuscenes is mentioned for the OOD datasets
    + in table 3, the meaning of dagger is not explained

- In general pretraining DINOv2 on automative data as it is, is not expected to provide good results as the content and image sizes are very different. See [k] for a discussion

- The t-sne plot does not look convincing as the two distributions are still quite separated

**References:**

[a] Bijelic et al., A Benchmark for Lidar Sensors in Fog: Is Detection Breaking Down?, arXiv 2019

[b] Bijleic et al., Seeing Through Fog Without Seeing Fog: Deep Multimodal Sensor Fusion in Unseen Adverse Weather, CVPR 2020

[c] Linnhoff et al., Simulating Road Spray Effects in Automotive Lidar Sensor Models, IEEE Sensors 2022

[d] Yang et al., Depth Anything V2, NeurIPS 2024

[e] Bochkovskii et al., Depth Pro: Sharp Monocular Metric Depth in Less Than a Second, ICLR 2025

[f] Sakaridis et al., ACDC: The adverse conditions dataset with correspondences for semantic driving scene understanding, ICCV 2021

[g] Franchi et al., Robust Semantic Segmentation with Superpixel-Mix, BMVC 2021

[h] Vu et al., The BRAVO Semantic Segmentation Challenge Results in UNCV2024, ECCV workshops 2024

[i] Sirko-Galuchenko et al., OccFeat: Self-supervised Occupancy Feature Prediction for Pretraining BEV Segmentation Networks, CVPR workshops 2024

[j] Jaritz et al., xMUDA: Cross-Modal Unsupervised Domain Adaptation for 3D Semantic Segmentation, CVPR 2020

[k] Chen et al., MultiSiam: Self-supervised Multi-instance Siamese Representation Learning for Autonomous Driving, ICCV 2021

**Questions:**

This paper takes an interesting direction of study: how to improve 2D representations by taking useful cues and signals from 3D representations of Lidar points clouds.

I find the endeavor of the authors nice, with a good story. However I do have several concerns regarding the novelty of the method, the chosen experimental settings and the limited to no comparison with relevant baselines in more standard tasks such as BEV perception, 3D object detection.

My current rating is leaning towards reject at this time, but I'm looking forward for the rebuttal.

Here are a few questions and suggestions that could be potentially addressed in the rebuttal or in future versions of this work (please note that suggested experiments are not necessarily expected to be conducted for the rebuttal):

1. Study the impact of stage 1 data split, but using stronger 3D encoder for stage 2, e.g., ScaLR trained on multiple datasets.

2. Inclusion of semantic segmentation evaluation on actual rain, fog, snow images. The model trained on Cityscapes can be directly tested on ACDC for that.

3. Comparison with DepthAnythingv2 on depth estimation tasks. What it DepthAnythingv2 was used in CD in place of DINOv2?

4. What if the 2D encoder and the 3D encoder are trained together without the alternate freezing, like in UniPad?

5. Extension of automotive experiments to BEV semantic segmentation and 3D object detection and comparison with relevant methods

---

> ### Author Response · Authors · 2025-11-24
> **Response to Reviewer jfki's Review**
>
> Response to Reviewer jfki's Review
> We thank the reviewer jfki for highlighting several strengths of our work, including the practical significance of improving perception robustness under challenging real-world conditions, the clarity and readability of the manuscript with its visual explanations, and the strong empirical validation across multiple datasets, tasks, and distribution shifts. We address the reviewer’s concerns and questions below.
>
> ---
>
> ## **Q1: Study the impact of stage 1 data split with multiple datasets**
>
> As the reviewer’s suggestion,
>
> We verify the scalability of our method by examining whether it can operate effectively on multiple datasets.
>
> First, we train the DINOv2 ViT-L model in Stage 2 using the ScaLR checkpoint, which is trained across multiple datasets (nuScenes, KITTI, Pandar64, and PandarGT) (Denoted by “Multiple”).
>
> Second, in Stage 1, we exclude adverse-weather-condition data (“denoted by Multiple-clear”)  and proceed with the same process.
>
> The below table confirms that our method is not limited to nuScenes and can benefit from multi-dataset training.
>
> Furthermore, the data-split strategy is also effective in the multi-dataset scenario.
>
> | Method   | Stage 1 Dataset   | Stage 2 Dataset | Seg. (mIoU ↑) |
> |----------|--------------------|------------------|----------------|
> | **DINOv2** | -                  | -                | 53.6           |
> | + CD     | Multiple           | Multiple         | 57.1           |
> | + CD     | Multiple-clear     | Multiple         | **57.5**       |
>
> Note that the Stage 1's result in the second row is obtained from the ScaLR official checkpoint.
>
> ---
>
> ## **Q2: Another test set that includes adverse weather conditions**
>
> We also train and evaluate our method on ACDC, a real adverse-weather dataset that includes night, rain, fog, and snow.
>
> The results below show that our approach remains effective on another real-world dataset with diverse adverse conditions.
>
> | Method   | Eval. data | Seg. (mIoU ↑) |
> |----------|------------|----------------|
> | **DINOv2** | Night      | 52.8           |
> | + CD     | Night      | **53.1**       |
> | **DINOv2** | Rain       | 60.7           |
> | + CD     | Rain       | **68.0**       |
> | **DINOv2** | Fog        | 66.4           |
> | + CD     | Fog        | **74.3**       |
> | **DINOv2** | Snow       | 64.5           |
> | + CD     | Snow       | **70.1**       |
>
>
>
> ---
>
> ## **Q3: What if DepthAnythingv2 was used in CD in place of DINOv2?**
>
> As requested, we validate this by applying our approach to DepthAnythingv2, a widely used MDE foundation model.
> DepthAnythingv2 follows an image-encoder + depth-decoder structure, so we use its pre-trained image encoder as the starting point for our method from the official checkpoint.
> For a fair comparison, both DINOv2 and DepthAnythingv2 use a ViT-B encoder and a linear depth head.
>
> As shown in the table below, our method further improves the performance of DepthAnythingv2 across both segmentation and depth estimation metrics.
>
> | Method              | Arch.      | Seg. (mIoU ↑)         |               | Depth. (RMSE ↓) |
> |---------------------|------------|------------------------|---------------|------------------|
> |                     |            | **nuScenes**           | **nuImages**  | **nuScenes**     |
> | DepthAnythingV2     | ViT-B/14   | 53.5                   | 75.1          | 7.32             |
> | + CD                | ViT-B/14   | **55.1**               | **75.8**      | **6.86**         |
>
> ---
>
> ## **Q4: What if the 2D encoder and the 3D encoder are trained together**
>
> As requested, we merge Stages 1 & 2 into a single joint training.
>
> With our distillation loss, this joint training easily falls into trivial solutions (e.g., feature collapse).
>
> Even when replacing the L2 loss with a contrastive loss and increasing the training epochs, the one-stage setup remains unstable and performs worse than our two-stage setup (see the table below).
>
> | Method  | 2-Stages | Epochs | Seg. (mIoU ↑) |
> |---------|----------|--------|----------------|
> | **DINOv2** | -        | -      | 49.2           |
> | + CD    | ❌       | 1      | 38.5           |
> | + CD    | ❌       | 5      | 30.1           |
> | + CD    | ❌       | 10     | 25.8           |
> | + CD    | ✅       | 1      | **51.9**       |
>
> This verifies the effectiveness of our two-stage design.

---

> ### Author Response · Authors · 2025-11-24
>
> ## **Q5: Bev-based downstream tasks**
>
> BEV-based methods typically require multi-view image inputs, specialized BEV-space lifting modules, and several task-specific components.
>
> These additional factors introduce many influences beyond the representation itself, making it difficult to isolate and assess the quality of the 2D representation.
>
> Since our goal is to evaluate the enhanced 2D representation in a controlled manner, we follow prior works that study or improve 2D representations (e.g., DINOv2, FiT3D, Condense) and therefore focus on monocular, linear-probing downstream tasks such as depth estimation and semantic segmentation, which are standard in this line of research.
>
> To further strengthen the evaluation, we also report results on monocular depth-aware video panoptic segmentation on automotive datasets in the main paper.
>
> ---
>
> ## **W1: Robustness of LiDAR & dataset assumption**
>
> We agree that LiDAR can also be degraded under heavy rain, fog, and snow.
>
> Our intention is not that LiDAR is invariant to adverse weather, but rather that it is relatively more robust than cameras, which is the aspect we leverage in our method.
>
>
> Physically, LiDAR uses near-infrared wavelengths, which scatter less than visible light, making it less sensitive than cameras under rain or fog [A1].
>
> In our work, we focus on this relative robustness and empirically show that LiDAR provides a more reliable signal under adverse conditions, serving as a relatively more stable supervisory signal for improving the 2D encoder.
>
> Regarding the composition of the LVD-142M dataset used for DINOv2 pre-training, we acknowledge that the dataset is private and its exact distribution is unknown.
>
> Empirically, we observe that DINOv2 is vulnerable to night-scene degradation.
>
> We will revise the wording in the paper to reflect both points more precisely and avoid overstatement.
>
> [A1] https://www.hesaitech.com/rain-and-fog-got-you-down-lidar-clears-the-way-for-safer-intelligent-driving/
>
> ---
>
> ## **W2: Practicality and real-world applicability**
>
> While our pre-training requires paired image–LiDAR data, the downstream stage uses only RGB images and the 2D encoder.
>
> Even without LiDAR at inference, we still observe improvements in OOD generalization, highlighting that our method has practical value for real-world deployment.
>
> ---
>
> ## **W3: Choice of evaluation protocol**
>
> We use nuScenes 2D semantic segmentation as an in-domain evaluation because our pre-training is also performed on nuScenes.
>
> To address the sparsity of LiDAR-projected labels, we additionally include nuImages, which provides dense annotations.
>
> We also evaluate widely adopted tasks in the community such as nuScenes depth estimation and standard OOD benchmarks, including those commonly used in DINOv2-based studies (e.g., DINOv2, FiT3D, Condense).
>
> ---
>
> ## **W4: Another baselines except DINOv2**
>
> Methods such as UniPAD, BEVDistill, DistillBEV, and OccFeat mainly target task-specific 3D perception settings (e.g., BEV detection/segmentation) and rely on multi-view inputs, BEV lifting modules.
>
> Our work instead focuses on improving a general 2D encoder—taking a single RGB image input—under a self-supervised representation-learning setting, so that it can benefit a broad range of downstream tasks both in-domain and out-of-domain.
>
> For this reason, we use DINOv2, a widely adopted 2D foundation encoder, as the primary baseline.
>
> ---
>
> ## **W5: Comparison with FiT3D and Condense**
>
> FiT3D and Condense, while not using LiDAR, still exploit 3D cues through Gaussian Splatting and NeRF, making them reasonably comparable methods.
>
> However, both methods are trained on indoor multi-view datasets that match their evaluation domain, whereas our method is trained solely on outdoor driving data.
>
> Despite this disadvantage, our outdoor method still outperforms them on out-of-domain indoor benchmarks, demonstrating strong generalization and the effectiveness of our approach.
>
> ---
>
> ## **W6: Distilling a Lidar encoder in image encoders has been explored**
>
> While prior works explore LiDAR-to-image distillation, they are typically designed for specific downstream tasks and do not aim to improve the robustness of a self-supervised 2D encoder.
>
> Our novelty is not merely splitting the data across stages.
>
> Our two-stage design is intentional and necessary:
>
> - Stage 1 (Pre-alignment) aligns the modalities so that LiDAR features can be used reliably as a teacher.
>
> - Stage 2 (3D-anchored self-supervision) injects 3D robustness while preserving the original capabilities of the 2D encoder.
>
> These stages play complementary roles and explicitly address robustness under noisy and adverse-weather conditions—an aspect not handled in previous cross-modal distillation frameworks.
>
> Thus, CD provides a new formulation for cross-modal representation learning, using calibrated 3D signals to enhance a self-supervised 2D foundation model—something prior work has not explored.

---

> > ### Author Response · Authors · 2025-11-24
> >
> > ## **Minor Misc.**
> >
> > **Typos.**
> > Thank you for pointing out the typos. We have corrected them accordingly.
> >
> > **Discussion about [k].**
> > Since DINOv2 is not pre-trained on automotive data, a further-trained variant, DINOv2+, is introduced for a fair comparison with ours.
> >
> > Additionally, the proposed method is not limited to DINOv2, and its effectiveness is also demonstrated on CLIP and Depth-Anything v2, which are likely to include automotive-related scenes in their training data.
> >
> > **t-SNE.**
> > Compared with DINO and DINOv2+, our features exhibit relatively more overlap, which we believe indicates the intended feature distribution.

---

> > > ### Comment · Reviewer_jfki · 2025-11-27
> > >
> > > I would like to thank the authors for the detailed and informative rebuttal. I imagine they invested significant effort into clarifying the concerns from the 4 reviewers and I'm confident they will improve this work. I've read the other reviews and the responses of the authors to them and skimmed through the paper again.
> > >
> > >
> > > Here are some follow-up comments and questions:
> > >
> > > **Q2: ACDC evaluation**
> > > - the authors might have  misunderstood suggestion about the ACDC and corrupted dataset evaluations. They train on different ACDC splits of weather and then evaluate on the validation splits of the same conditions.
> > > - This is not what I suggested and not quite an OOD evaluation.
> > > - I suggest to train the model as before on Cityscapes and evaluate on the validation splits of ACDC, which is the more common practice for OOD evaluation and much cheaper to conduct as there is no need to finetune the model on different data splits.
> > >
> > >
> > > **Q3: Comparison with DAv2**
> > > - I'm not sure I understand what results the author are reporting. Could they please clarify?
> > > - Are they comparing DAv2 + LP with DAv2 + CD + LP or with DINOv2 + CD + LP?
> > > - It's a surprising and interesting result if DINOv2 + CD + LP outperforms DAv2 + LP. What is the typical standard deviation on the scores?
> > >
> > >
> > > **Q5: Other automotive downstream tasks**
> > > - 3D object detection is an important task for automotive applications and would highly benefit from depth priors as argued by the proposed approach. Besides 3D object detection can be done in a monocular way without multi-camera or BEV.
> > > - Sparse semantic segmentation is not common in practical applications. nuImages results are however useful.
> > >
> > >
> > > **W3: Evaluation protocols**
> > > - nuscenes is used in this work as workhorse for main results and ablations. The proposed evaluation protocol remains limited and less common as the semantic segmentation is evaluated only on the sparse point cloud points projected in the image space. This could lead to fluctuations in reported scores.
> > > - The authors argue that they additionally include nuImages results with dense annotations. While nuImages is interesting, it is however not very dense either, as it has labels for road, vehicles and road users only.
> > >
> > >
> > > **W5: Comparison with FiT3D and Condense**
> > > - I disagree with the authors, 3D cues from NeRF or Gaussian Splatting are not comparable with precise 3D information fron Lidar point clouds.
> > > - I still consider that the comparison is not apples-to-apples and even so the comparison against these models is conducted only in OOD settings.
> > > - For in-domain evaluation only the original encoder (e.g., DINOv2) is used as baseline.

---

> ### Author Response · Authors · 2025-12-03
>
> ## **Q2: ACDC evaluation**
>
> Thank you for the clarification.
>
> Following the suggestion, we evaluate the model trained on Cityscapes directly on the ACDC validation splits without any training (See the below table).
>
> The updated table shows consistent improvements across all adverse weather conditions, demonstrating strong robustness of our method.
>
> | Method   | Train. data | Eval. data | Seg. (mIoU ↑) |
> |----------|-------------|------------|----------------|
> | **DINOv2** | ACDC Night | ACDC Night | 52.8 |
> | + CD     | ACDC Night | ACDC Night | **53.1** |
> | **DINOv2** | ACDC Rain  | ACDC Rain  | 60.7 |
> | + CD     | ACDC Rain  | ACDC Rain  | **68.0** |
> | **DINOv2** | ACDC Fog   | ACDC Fog   | 66.4 |
> | + CD     | ACDC Fog   | ACDC Fog   | **74.3** |
> | **DINOv2** | ACDC Snow  | ACDC Snow  | 64.5 |
> | + CD     | ACDC Snow  | ACDC Snow  | **70.1** |
> | **DINOv2** | CityScapes | ACDC Night | 46.4 |
> | + CD     | CityScapes | ACDC Night | **49.0** |
> | **DINOv2** | CityScapes | ACDC Rain  | 63.1 |
> | + CD     | CityScapes | ACDC Rain  | **66.9** |
> | **DINOv2** | CityScapes | ACDC Fog   | 69.9 |
> | + CD     | CityScapes | ACDC Fog   | **72.1** |
> | **DINOv2** | CityScapes | ACDC Snow  | 62.3 |
> | + CD     | CityScapes | ACDC Snow  | **66.7** |
>
> ---
>
> ## **Q3: Comparison with DAv2**
>
> Thank you for insightful comment.
>
> In the previously reported table, we compared DAv2 + LP with DAv2 + CD + LP to show that our approach is also effective when applied to DAv2.
>
> When we place DINOv2 + LP and DINOv2 + CD + LP side by side,
> DINOv2 + CD + LP outperforms DAv2 + LP across all metrics, with especially notable gains in depth estimation (See the below table).
>
> | Method              | Arch.      | Seg. (mIoU ↑)         |               | Depth. (RMSE ↓) |
> |---------------------|------------|------------------------|---------------|------------------|
> |                     |            | **nuScenes**           | **nuImages**  | **nuScenes**     |
> | DINOv2     | ViT-B/14   | 52.3                   | 74.9          | 8.01             |
> | + CD                | ViT-B/14   | **55.5**               | **76.4**      | **7.18**         |
> | DepthAnythingV2     | ViT-B/14   | 53.5                   | 75.1          | 7.32             |
> | + CD                | ViT-B/14   | **55.1**               | **75.8**      | **6.86**         |
>
> The results indicate that applying our method to a self-supervised pre-trained image encoder such as DINOv2 can surpass even a depth-specialized foundation model in depth estimation.
>
> The results also demonstrate that our method can further improve depth-specialized foundation models themselves.
>
> As the reviewer mentioned, we also find this result both surprising and exciting. We believe this finding provide a proper reference point for future research.
>
> Depth estimation shows a standard deviation of ≈ ±0.01, and semantic segmentation shows ≈ ±0.04 mIoU across three runs, indicating stable improvements.
>
> ---
>
> ## **Q5: Other automotive downstream tasks**
>
> Regarding the point on other automotive downstream tasks, we conduct an additional experiment on a camera-based 3D object detection task to further assess the 3D awareness of the proposed method.
>
> The PETR [A2] decoder remains trainable while the image backbone is replaced with ViT-S pre-trained by DINOv2, FiT3D, or our method and then frozen.
>
> Training and evaluation are performed on nuScenes, following standard 3D-detection metrics (mAP and NDS), and all experiments are conducted using the MMDetection3D [A3] repository.
>
> As shown in the table, FiT3D performs worse than DINOv2, whereas the proposed method consistently improves over DINOv2 on both metrics.
>
> This result strengthens the evidence that the proposed method enhances 3D-aware 2D representations and benefits practical automotive tasks such as camera-based 3D object detection.
>
> | Method    | Decoder | mAP ↑ | NDS ↑ |
> |-----------|---------|--------|--------|
> | **DINOv2** | PETR       | 10.6   | 19.0   |
> | + FiT3D   | PETR    | 6.1    | 16.2   |
> | + CD      | PETR       | **11.5** | **19.2** |
>
> [A2] Liu, Yingfei, et al. "Petr: Position embedding transformation for multi-view 3d object detection." European conference on computer vision. Cham: Springer Nature Switzerland, 2022.
>
> [A3] https://github.com/open-mmlab/mmdetection3d

---

> ### Author Response · Authors · 2025-12-03
>
> ## **W3: Evaluation protocols**
>
> On nuScenes semantic segmentation evaluation, the score fluctuations are small.
> Across three runs, semantic segmentation shows a low deviation of ≈ ±0.04 mIoU on linear probing and ≈ ±0.18 mIoU on fine-tuning, indicating that the improvements are stable and not driven by fluctuations.
>
> In addition, semantic segmentation on nuImages is not limited to only road, vehicles, and road users.
> It includes 11 classes such as barrier, bicycle, bus, car, construction vehicle, motorcycle, pedestrian, traffic cone, truck, trailer, and drivable surface, providing dense labels.
>
> For ablation studies where only nuScenes semantic segmentation was shown (Table B, D, and K), we add nuScenes depth estimation results to these tables.
>
> The trend remains unchanged, which strengthens the reliability of the ablations.
> We appreciate that this point helps improve the reliability of our ablation studies.
>
> ---
>
> ## **W5: Comparison with FiT3D and Condense**
>
> Our comparison with FiT3D and Condense focuses on methods that improve the 2D representations of models like DINOv2.
>
> These works aim to enhance 3D awareness of 2D representations, which makes them the closest to our scope.
>
> To the best of our knowledge, there are no prior works that improve self-supervised pre-trained 2D representations (e.g., DINOv2) using LiDAR data.
>
> Although a fully apples-to-apples comparison is difficult, we conduct additional experiments to address this concern and to ensure a consistent use of training data.
>
> For this purpose, we consider BEVDistill and DistillBEV, which meet the following conditions:
>
> • they use both LiDAR and image data,
>
> • their image encoders are trained.
>
> Both BEVDistill and DistillBEV train on nuScenes (in-domain) for 3D object detection using LiDAR and images, and their image encoders (ResNet50) are updated during training.
>
> We linear probe their officially released image encoder weights on nuScenes semantic segmentation and depth estimation in an in-domain setting.
>
> For a fair comparison, we also linear probe ResNet50 pre-trained by DINO and the same backbone further trained by our method. (Note that there are no ResNet50 pre-trained by DINOv2)
>
> The results in the table show that our method brings larger performance gains on downstream tasks, even compared to methods that use LiDAR data (See the below table).
>
> | Method       | Arch.     | Seg. (mIoU ↑) | Depth (RMSE ↓) |
> |--------------|-----------|----------------|-----------------|
> | **DINO** | ResNet50 | 38.4 | 9.01 |
> | + CD         | ResNet50 | **46.1** | **7.83** |
> | **BEVDistill** | ResNet50 | 36.1 | 9.93 |
> | **DistillBEV** | ResNet50 | 44.9 | 9.11 |
>
> ---
>
> ## **W5: In-domain evaluation with FiT3D and Condense**
>
> FiT3D and Condense are already evaluated in the in-domain setting.
>
> Appendix B.9 (Table I) reports these results, where our method achieves higher performance on all metrics.

---

### Author Response · Authors · 2025-11-24
**Revision Summary**

We thank all reviewers — $\textcolor{#F1433F}{Reviewer \ jfki}$, $\textcolor{#E7B500}{Reviewer \ hsWC}$, $\textcolor{#A9CF54}{Reviewer \ KQwi}$, $\textcolor{#70B7BA}{Reviewer \ fJFr}$  — for their thoughtful and constructive feedback.

We are encouraged that the reviewers collectively highlighted the importance and practicality of our problem setting for real-world robustness ($\textcolor{#F1433F}{jfki}$, $\textcolor{#A9CF54}{KQwi}$), appreciated the clarity and simplicity of the presentation as well as the effective use of complementary sensor properties ($\textcolor{#E7B500}{ hsWC}$), and acknowledged the strength of our empirical evaluation, including clear visualizations and consistent improvements across tasks and settings ($\textcolor{#F1433F}{jfki}$, $\textcolor{#A9CF54}{KQwi}$, $\textcolor{#70B7BA}{fJFr}$).


We summarize the revision as follows:
- Clarifying robustness of LiDAR & dataset assumption $->$ Introduction on page 2 ($\textcolor{#F1433F}{jfki}$)
- Cite related works $->$ Related work on page 2-3 ($\textcolor{#F1433F}{jfki}$, $\textcolor{#A9CF54}{KQwi}$)
- “Stage 1” and “Stage 2” for readability $->$ Fig 2 on page 3 ($\textcolor{#A9CF54}{KQwi}$)
- Adding real adverse weather test set results $->$ Section B.3 and Table C on appendix ($\textcolor{#F1433F}{jfki}$, $\textcolor{#E7B500}{hsWC}$)
- Adding DepthAnythingV2 results $->$ Section B.1 and Table A on appendix ($\textcolor{#F1433F}{jfki}$, $\textcolor{#E7B500}{hsWC}$)
- Adding LiDAR corruption results $->$ Section B.2 and Table B on appendix ($\textcolor{#E7B500}{hsWC}$, $\textcolor{#A9CF54}{KQwi}$)
- Adding with and without 2-Stages results $->$ Section B.11 and Table K on appendix ($\textcolor{#F1433F}{jfki}$, $\textcolor{#A9CF54}{KQwi}$)
- Adding multiple datasets results $->$ Section B.4 and Table D on appendix ($\textcolor{#F1433F}{jfki}$)
- Adding comparison with LiDAR-based methods $->$ Section B.5 and Table E on appendix ($\textcolor{#F1433F}{jfki}$)
- Adding camera-based 3D object detection results $->$ Section B.6 and Table F on appendix ($\textcolor{#F1433F}{jfki}$)
- Adding computational overhead table $->$ Section C.1 and Table J on appendix  ($\textcolor{#A9CF54}{KQwi}$)

We sincerely appreciate the reviewers’ thoughtful feedback, which helps us further strengthen and clarify our submission.

---

> ### Author Response · Authors · 2025-12-03
> **Summary of Rebuttals for the Area Chair**
>
> Dear Area Chair,
>
> In light of the unusual reviewing situation, we would like to provide a brief summary specifically to assist your decision, complementing our earlier “Revision Summary” and detailed rebuttals.
>
> ---
>
> ## 1. All weaknesses and questions from the four reviewers are fully addressed
>
> Throughout the rebuttal, every weakness and every question is answered without omission.
>
> Among them, several points repeatedly arise across multiple reviews and contribute to improving the work.
>
> These key issues are addressed through the following three additions:
>
> ### (1) Real adverse-weather OOD evaluation (jfki, hsWC)
>
> A real adverse-weather OOD evaluation is added.
>
> The ACDC results show that our method consistently improves DINOv2 across night, rain, fog, and snow, demonstrating strong robustness under real adverse-weather conditions.
>
> ### (2) DepthAnything v2 comparison and integration (jfki, hsWC)
>
> We examine our method's generalizable adaptability to DepthAnythingV2, a widely used MDE foundation model.
>
> Our method (CD) improves DepthAnythingV2 itself, and DINOv2+CD surpasses DAv2, indicating that applying CD to a self-supervised image encoder such as DINOv2 can outperform even a depth-specialized foundation model.
>
> ### (3) Stability under LiDAR corruption (hsWC, KQwi)
>
> We conduct LiDAR corruption experiments using Gaussian noise and density reduction.
>
> These experiments confirm that our method maintains stable improvements even under moderate LiDAR degradation.
>
> ## 2. How the reviewers’ impressions would have changed
>
> After revisiting all reviews, responses, and the paper, reviewer jfki provides the following follow-up message:
>
> “I would like to thank the authors for the detailed and informative rebuttal… I’m confident they will improve this work.”
>
> This statement appears after careful re-reading of the discussion and suggests a notably improved impression of the work.
>
> All follow-up questions and comments are also fully answered with additional experiments and clarifications.
>
> Although the reviewer scores remain the same due to the unusual reviewing situation, we hope that the Area Chair will take into account that reviewer jfki’s impression of the work clearly changed in a positive way after the rebuttal.
>
> ---
>
> Thank you sincerely for the time and effort dedicated to assessing this submission during a challenging review cycle.
>
> Sincerely,
> The Authors

---

### Meta-Review · Area_Chair_pe2S · 2025-12-28

**Summary:**

This paper introduces Collaborative Distillation (CD), a self-supervised method to improve the robustness of pretrained 2D image encoders to adverse weather (e.g., rain, low-light, night, corruptions). Nearly all reviewers tend to recommend rejection for this submission due to varisous concerns. While the authors argue that 3D Lidar representations are highly robust to adverse weather conditions compared to 2D image representations, this might not be true sometimes. The evaluation protocol is somewhat limited and less common. The comparison is not fair given the geometric information is not the same. Also, the paper primarily combines existing techniques into a new use case, which seems more engineering-oriented.

**Reviewer Scores:**

NA

---

### Decision · Program_Chairs · 2026-01-26

Reject